# Scientific Logicality Enriched Methodology for LLM Reasoning: A Practice in Physics

**Zhaoxin Yu** [* 1 2]  **Nan Xu** [* 1 3]  **Kun Chen** [2 1]  **Jiahao Zhao** [1 3]  **Lei Wang** [3]  **Wenji Mao** [1 2]

## Abstract

With the continuous advancement of reasoning abilities in Large Language Models (LLMs), their application to scientific reasoning tasks has gained significant research attention. Current research primarily emphasizes boosting LLMs' performance on scientific QA benchmarks by training on larger, more comprehensive datasets with extended reasoning chains. However, these approaches neglect the essence of the scientific reasoning process—logicality, which is the rational foundation to ensure the validity of reasoning steps leading to reliable conclusions. In this work, we make the first systematic investigation into the internal logicality underlying LLM scientific reasoning, and develop a scientific logicality-enriched methodology, including a set of assessment criteria and data sampling methods for logicality-guided training, to improve the logical faithfulness as well as task performance. Further, we take physics, characterized by its diverse logical structures and formalisms, as an exemplar discipline to practise the above methodology. For data construction, we extract scientific problems from academic literature and sample a high-quality dataset exhibiting strong logicality. Experiments based on three different backbone LLMs reveal that: 1) the training data we constructed can effectively improve the scientific logicality in LLM reasoning; and 2) the enriched scientific logicality plays a critical role in solving scientific problems. Code is available at https://github.com/ScienceOne-AI/PhysLogic.

---

[*]Equal contribution [1]State Key Laboratory of Multimodal Artificial Intelligence Systems, Institute of Automation, Chinese Academy of Sciences, Beijing, China [2]School of Artificial Intelligence, University of Chinese Academy of Sciences, Beijing, China [3]Beijing Wenge Technology Co., Ltd, Beijing, China. Correspondence to: Wenji Mao <wenji.mao@ia.ac.cn>.

*Proceedings of the 43rd International Conference on Machine Learning*, Seoul, South Korea. PMLR 306, 2026. Copyright 2026 by the author(s).

## 1. Introduction

With the continuous advancement of Large Language Models (LLMs), significant research efforts and progress have been made to apply them to solving scientific problems across disciplines such as mathematics, physics, and chemistry, aiming to enhance the efficiency in academic research and education (Zhang et al., 2024b; Zheng et al., 2025b). For complex problem solving, early work focuses on strategies at inference time and designs structured procedures that guide LLMs to reason step by step (Wei et al., 2022; Wang et al., 2023). More recently, reasoning models such as DeepSeek R1 and OpenAI o1 adopt a training-time paradigm that instills sophisticated reasoning abilities during learning (Guo et al., 2025; Jaech et al., 2024), yielding strong performance across disciplinary reasoning tasks (Hu et al., 2025). Building on this paradigm, a number of studies have constructed training corpora containing long and complex scientific reasoning traces to train LLMs (Yuan et al., 2025; Fan et al., 2025; Zhang et al., 2024a; Lu et al., 2025). Meanwhile, many benchmarks are built to evaluate models' scientific problem-solving capability by formulating question-answer (QA) tasks in diverse formats (Rein et al., 2024; Wang et al., 2024).

However, these studies narrowly cast scientific reasoning as an end-to-end natural language processing task and neglect the essence of the scientific reasoning process–**logicality**, which encompasses a set of interrelated concepts, methods and principles, and forms the rational foundation that ensures the validity of reasoning steps and the reliability of conclusions (Popper, 2005; Díaz et al., 2023). Figure 1 illustrates an example of the reasoning paradigms of DeepSeek-R1 and a professional human in answering a scientific question, where humans typically follow a series of interconnected logical steps including *problem formalization*, *model generation*, *evidence generation*, *evidence evaluation* and *drawing conclusions*, etc. (corresponding to the epistemic activities in Fischer et al. (2014)). A related study reveals that each scientific discipline has its own paradigm of reasoning, which is the way of solving problems that are generally held in common by the community of those practising in this discipline (Dowden, 1993). In contrast, the reasoning traces generated by current reasoning LLMs are often an ad

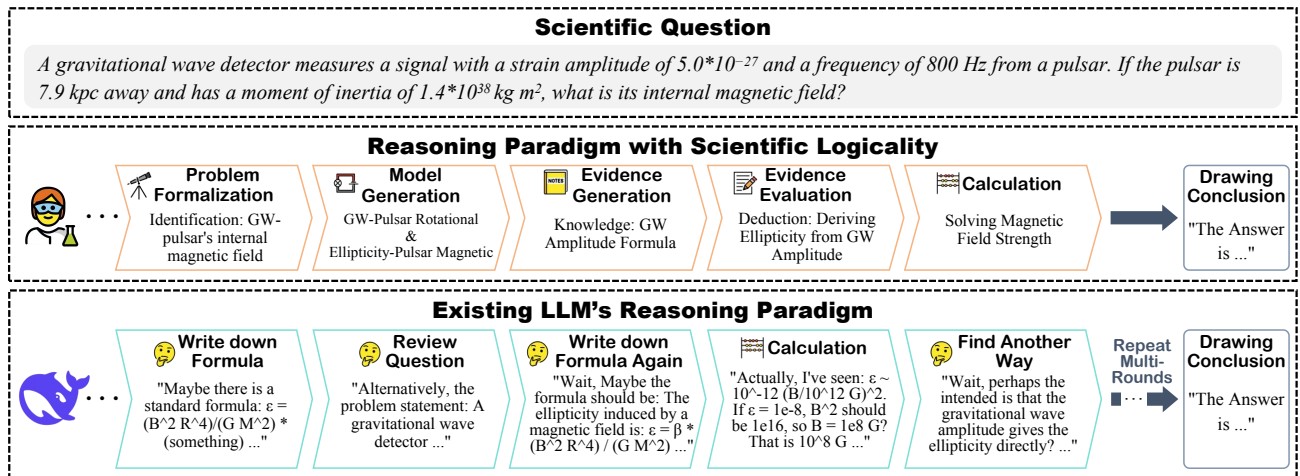

*Figure 1.* Comparison of the scientific reasoning paradigms between DeepSeek-R1 and a professional (human): LLM lacks the scientific logicality possessed by human experts.

hoc aggregation of recall, review, and self-reflection steps with lengthy iterations and relatively weak logical coherence between them.

In this paper, we conduct the first systematic investigation into the internal logicality underlying LLM scientific reasoning. First, we design a set of criteria with three dimensions: logical fidelity, causal connection, and inferential progress to assess scientific logicality during the reasoning process; then we design two SFT data sampling methods, based on distillation and reasoning style transfer, respectively, to enhance scientific logicality in LLM reasoning. To practice the above methodology, we choose physics as an exemplar discipline, whose reasoning paradigm spans formal derivation and computation in formal sciences (e.g., pure math), as well as real-world modeling and experimental methodology in natural sciences. More concretely, we construct a set of high-quality QA datasets extracted from the core logical derivations of physics papers, from which we sample 80k SFT instances and 864 benchmark examples. Both in-domain and out-of-domain experiments are conducted to examine the effect of SFT on enhancing LLMs' scientific reasoning logicality and their final task performances.

The contributions of our work are summarized as follows:

1. We make the first exploration of the logicality in LLM scientific reasoning, and design logicality-centric assessment criteria and data sampling methods to improve LLMs' scientific reasoning process and performance. Empirical studies involving third-party verification fully demonstrate the validity of the criteria.

2. We construct a high-quality QA dataset extracted from physics papers and on this basis, build the PHYSLOGIC benchmark, the first of its kind for systematically evaluating the logicality of LLM physics reasoning, together

with two distinct logicality-enriched training datasets.

3. We conduct extensive experiments and the results on both PHYSLOGIC benchmark and three representative public benchmarks show that our constructed training dataset can effectively improve LLM logicality in physics reasoning and the final task performances.

## 2. Methodology

Scientific Reasoning is regarded as the cognitive processes required to use the scientific method, consisting of *a series of steps* (Díaz et al., 2023), which are aligned to the epistemic definition of Fischer et al. (2014). Thus, solving a scientific problem involves distinct reasoning steps (we term them as *logical nexuses* [1][2]), denoted as $\mathcal{N} = \{\nu_1, \cdots, \nu_n\}$, where $n$ is the number of nexuses. Based on Fischer et al. (2014), logical nexuses (characterized by epistemic activities) might differ substantially in the relative weights in a discipline. These weights corresponding to $\mathcal{N}$ are denoted as $\mathcal{W} = \{w_1, \cdots, w_n\}$. The reasoning process of a problem solver is represented by a sequence of sentences $\mathcal{R} = \{r_1, \cdots, r_m\}$. Specifically, to ensure that each segment is semantically independent and complete while maintaining computational efficiency, we adopt a rule-based sentence-level segmentation scheme, a design choice that is widely adopted in prior work (Lightman et al., 2024; Sun et al., 2025; Macar et al., 2025). To enable quantitative assessment, we first encode these textual steps into vector representations. Using a sentence encoder, we transform the ground-truth nexuses $\mathcal{N}$ into embeddings $V_{\mathcal{N}} = \{v_{\nu_1}, \cdots, v_{\nu_n}\}$ and the reasoning steps $\mathcal{R}$ into em-

---

[1]https://www.merriam-webster.com/dictionary/nexus

[2]For specific examples, please refer to the data examples in Appendix J.

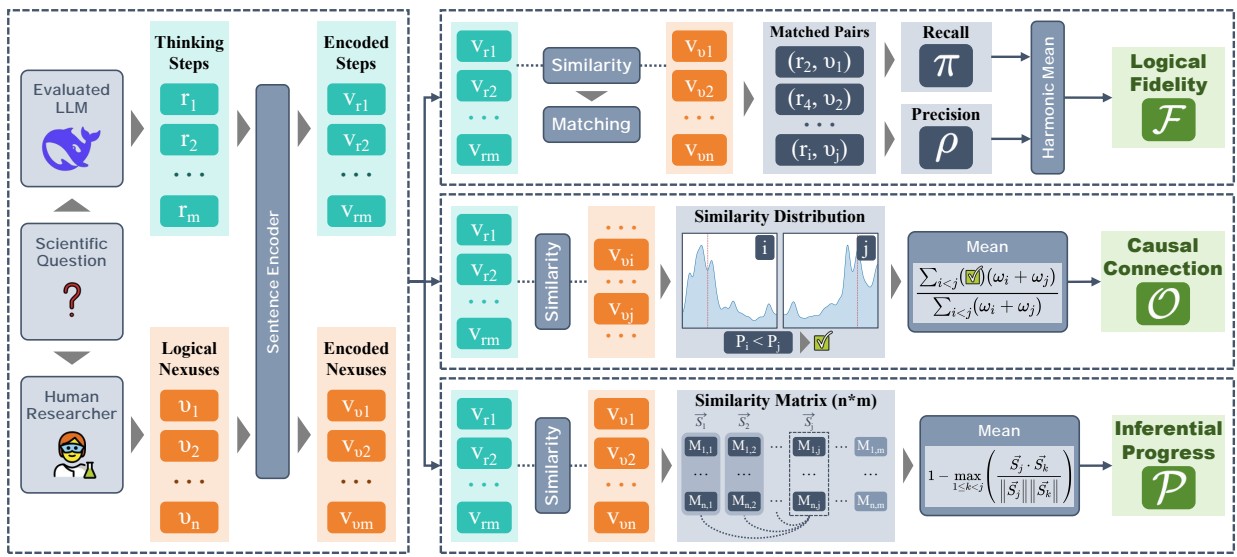

*Figure 2.* Assessment criteria for the scientific reasoning of LLMs, encompassing three dimensions: Logical Fidelity, Causal Connection, and Inferential Progress.

beddings $V_{\mathcal{R}} = \{v_{r_1}, \cdots, v_{r_m}\}$. In this chapter, we first propose multi-dimensional assessment criteria that use the nexus embeddings $V_{\mathcal{N}}$ as the ground truth to assess the scientific logicality of the reasoning process embeddings $V_{\mathcal{R}}$. Furthermore, given a dataset of scientific problems, where each entry comprises a QA pair, $\mathcal{N}$, and $\mathcal{W}$, we design two distinct logic-aware data sampling methods for SFT.

## 2.1. Assessment for Scientific Logicality in LLM Reasoning

As shown in Figure 2, we designed criteria with three complementary dimensions to assess the scientific logicality of an LLM's reasoning process:

**Logical Fidelity** $\mathcal{F}$ This metric quantifies the content alignment between the reasoning process under evaluation and the logical nexuses. We assess logical fidelity by aligning ground-truth logical nexus embeddings ($V_{\mathcal{N}}$) with the model's reasoning step embeddings ($V_{\mathcal{R}}$). First, we compute a cosine similarity matrix $M \in \mathbb{R}^{n \times m}$ between the two sets of embeddings. A greedy matching algorithm then identifies the optimal set of one-to-one pairs $\mathcal{C}$ by selecting matches that exceed a predefined similarity threshold $\tau$. Finally, we represent Logical Fidelity using the Logic F-Score ($\mathcal{F}$), which is the harmonic mean of the alignment's Logic Precision ($\pi$, which describes the proportion of the model's reasoning steps that are logically valid) and Logic Recall ($\rho$, which describes the proportion of logical nexuses that are covered by the model's reasoning):

$$\rho = \frac{\sum_{(i,j) \in \mathcal{C}} w_i \cdot M_{ij}}{\sum_{k=1}^{n} w_k}, \quad \pi = \frac{|\mathcal{C}|}{m}, \quad \mathcal{F} = 2 \cdot \frac{\pi \cdot \rho}{\pi + \rho}$$

where $w_i$ is the importance weight of nexus $\nu_i$, $n = |\mathcal{N}|$,

and $m = |\mathcal{R}|$. An $\mathcal{F}$ score of 1 indicates a perfect match with the logical nexuses, and higher values reflect a greater degree of content-level consistency between the model's reasoning and the logical nexuses.

**Causal Connection** $\mathcal{O}$ This dimension considers whether the LLM preserves the correct ordering between pairs of logical nexuses that have an inherent causal or derivational direction. When the model touches on both nexuses during reasoning, we examine whether the order it presents is consistent with the ground truth. This consistency is determined based on the relative distribution of semantic similarities. Specifically, for each nexus $\nu_i$, we compute its Positional Centroid $P_i$-its semantic center of mass within the model's reasoning process $\mathcal{R}$. The score $\mathcal{O}$ is the weighted proportion of nexus pairs that maintain their correct relative temporal order:

$$P_i = \frac{\sum_{j=1}^{m} j \cdot M_{ij}}{\sum_{j=1}^{m} M_{ij}}, \quad \mathcal{O} = \frac{\sum_{i<k \text{ s.t. } P_i < P_k} (w_i + w_k)}{\sum_{i<k} (w_i + w_k)}$$

An $\mathcal{O}$ score of 1 indicates a perfectly ordered sequence, while a score near 0.5 suggests random ordering.

**Inferential Progress** $\mathcal{P}$ This metric assesses whether the reasoning exhibits overall forward logical progression. For example, if the LLM repeatedly circles back to previously covered propositions or oscillates between them without making forward progress, the score on this dimension decreases. Specifically, it assesses reasoning efficiency by identifying non-productive patterns like conceptual loops. It analyzes the conceptual trajectory of the reasoning process, which is represented by a sequence of Similarity Vectors $[\vec{S_1}, \ldots, \vec{S_m}]$. Each vector $\vec{S_j}$ captures the similarity of a

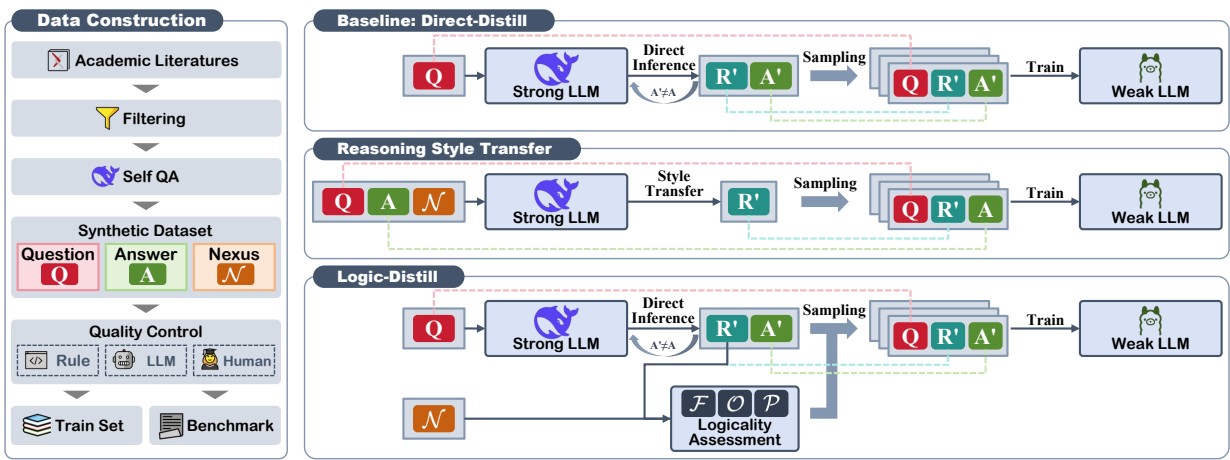

*Figure 3.* A pipeline to construct scientific QA data from academic papers, along with three SFT data sampling methods: a baseline and two comparative methods enriched with scientific logic.

reasoning step $r_j$ to all $n$ ground-truth nexuses:

$$\vec{S}_j = \begin{bmatrix} M_{1j} & M_{2j} & \dots & M_{nj} \end{bmatrix}^T$$

The final score $\mathcal{P}$ is the average conceptual novelty across the reasoning path, where the novelty of each step is one minus its maximum cosine similarity to any preceding step's vector:

$$\mathcal{P} = \frac{1}{m-1} \sum_{j=2}^{m} \left( 1 - \max_{1 \leq k < j} \left( \frac{\vec{S}_j \cdot \vec{S}_k}{\|\vec{S}_j\| \|\vec{S}_k\|} \right) \right)$$

A score close to 1 signifies a highly efficient, forward-progressing reasoning path, whereas a low score indicates significant conceptual repetition.

## 2.2. Scientific Logicality-Guided Data Sampling for SFT

To enhance the logicality of LLMs for scientific reasoning, we propose two logic-aware data sampling methods for SFT. These methods are designed for datasets where each entry consists of a question $Q$, an answer $A$, and a set of logical nexuses $\mathcal{N}$ with corresponding weights $\mathcal{W}$. Both approaches are illustrated in Figure 3.

**Reasoning Style Transfer** This method uses a powerful reasoning LLM ($\mathcal{L}$) in a style transfer task to generate a fluent reasoning path from the discrete logical nexuses. The model is prompted with the question $Q$ and the weighted nexuses ($\mathcal{N}$, $\mathcal{W}$) to synthesize a cohesive, narrative-style reasoning process. This effectively translates the structured logic into a natural thinking format. The operation is formalized as:

$$R' = \mathcal{L}(Q, \mathcal{N}, \mathcal{W})$$

where $R'$ is the synthesized reasoning. The final data entry for SFT is constructed as the tuple $\{Q, R', A\}$, pairing the synthesized reasoning with the original question and answer.

The reasoning process $R'$ is explicitly demarcated by "*think*" tags.

**Logical-Distillation** This strategy distills high-quality data by filtering the native reasoning of a powerful LLM ($\mathcal{L}$), using the ground-truth nexuses as an indirect supervisory signal.

First, the LLM is prompted with a question $Q$ to generate a native reasoning path $R'$ and answer $A'$:

$$(R', A') = \mathcal{L}(Q)$$

The generated reasoning $R'$ is segmented into discrete steps $\mathcal{R}$. We then assess this sequence against the ground-truth nexuses $\mathcal{N}$ using our evaluation suite to obtain scores for logic precision ($\pi$), logic recall ($\rho$), Causal Connection ($\mathcal{O}$), and Inferential Progress ($\mathcal{P}$).

To make the metrics comparable, we z-normalize each raw metric $X$ using the mean $\mu_X$ and standard deviation $\sigma_X$ computed over the full dataset $D_{\text{full}}$, and then apply a sigmoid to obtain a bounded score $\tilde{X} \in (0, 1)$, i.e., $\tilde{X} = \text{sigmoid}((X - \mu_X)/\sigma_X)$. We then compute the final logical score $\mathcal{S}$ as a weighted combination of logical fidelity and two auxiliary criteria[3]. Specifically, logical fidelity is defined as the harmonic mean of the normalized precision $\tilde{\pi}$ and recall $\tilde{\rho}$:

$$\mathcal{S} = \delta_{\mathcal{F}} \cdot \frac{2\tilde{\pi}\tilde{\rho}}{\tilde{\pi} + \tilde{\rho}} + \delta_{\mathcal{O}} \cdot \tilde{\mathcal{O}} + \delta_{\mathcal{P}} \cdot \tilde{\mathcal{P}}.$$

The final data entry for SFT is constructed as the tuple $\{Q, R', A'\}$. From $D_{\text{full}}$, we sample a subset $D$ by selecting instances in the top-$\kappa$ percentile according to $\mathcal{S}$:

$$D = \text{Top}_{\kappa}(D_{\text{full}}, \text{key} = \mathcal{S}).$$

---

[3]We report the weight settings and sensitivity analysis in Appendix G.

*Table 1.* Statistics for the final 80k instruction-tuning dataset

| Task Type | Data Number | $Q$ Tokens | $A$ Tokens | $\mathcal{N}$ Length | $\mathcal{N}$ Tokens | $\mathcal{R}$ Tokens | $\mathcal{R}_{\mathbf{RST}}$ Tokens |
|---|---|---|---|---|---|---|---|
| MCP | 12587 | 158.46 | 337.61 | 8.42 | 20.98 | 7971.93 | 902.82 |
| Comp. (E) | 17634 | 222.22 | 345.37 | 9.38 | 21.09 | 10757.04 | 1025.93 |
| Comp. (N) | 48005 | 219.86 | 355.41 | 8.90 | 22.86 | 9920.15 | 1137.26 |
| Proof | 1774 | 216.11 | 417.68 | 9.17 | 22.41 | 10518.05 | 1167.08 |

\* MCP: Multiple Choice Problem; Comp. (E): Expression Computation; Comp. (N): Numeric Computation; Proof: Proof-based Problem.

*Table 2.* Comparison of our proposed PHYSLOGIC benchmark with existing science benchmarks that include physics: Ours is the first benchmark to incorporate multiple, complementary dimensions for assessing the logicality of the reasoning process.

| Benchmark | Discipline | Difficulty Levels | Question Types | Answer Verification | Reasoning Verification | | |
|---|---|---|---|---|---|---|---|
| | | | | | Steps | Order | Progress |
| GPQA (Rein et al., 2024) | General | Grad. | MCQ | ✓ | ✗ | ✗ | ✗ |
| SciBench (Wang et al., 2024) | General | UG | Comp. | ✓ | ✗ | ✗ | ✗ |
| UGPhysics (Xu et al., 2025) | Physics | UG | MCQ, Comp. | ✓ | ✗ | ✗ | ✗ |
| PHYSICS (Feng et al., 2025) | Physics | Grad. | Comp. | ✓ | ✗ | ✗ | ✗ |
| PhysReason (Zhang et al., 2025) | Physics | HS | Comp. | ✓ | ✓ | ✗ | ✗ |
| PRISM-PHYSICS (Zhao et al., 2025) | Physics | UG, Grad. | Comp. | ✓ | ✓ | ✓ | ✗ |
| **PhysLogic** | Physics | HS, UG, Grad. | MCQ, Comp., Proof | ✓ | ✓ | ✓ | ✓ |

\* HS (High School), UG (Undergraduate), Grad. (Graduate), MCQ (Multiple-Choice Question), Comp. (Computational), Proof (Proof-based).

For comparison, we also use the full dataset $D_{\text{full}}$ as a baseline, directly distilling on the entire question dataset.

## 3. Data Foundation

**Dataset Construction** We instantiate our methodology in physics and build both training and evaluation data directly from research papers, which naturally encode rigorous deductive chains. We first collect 380,678 physics papers from arXiv and peer-reviewed journals, then use `DeepSeek-R1`[4] (Guo et al., 2025) (hereinafter "R1"; prompts in Appendix I.5) to retain theory-centric works and filter out reviews, empirical studies, and tool papers, yielding 118,039 papers. For each retained paper, we run a multi-turn dialogue with R1 to construct scientific problems: R1 generates a question $Q$ of specified type and difficulty from the derivations (with a 15:85 ratio of multiple-choice to open-ended questions), produces the solution in the form of a reasoning trajectory $R$ and, when applicable, a final answer $A$, and then extracts core logical nexuses $\mathcal{N}$ together with importance weights $\mathcal{W}$. We treat $(\mathcal{N}, \mathcal{W})$ as the logical gold standard for the problem. During the distillation process, to ensure that the distilled answer $A'$ matches the original answer $A$, we apply rejection sampling with a maximum of 5 retries.

**Quality Control** To guarantee the quality of the synthesized data, we designed 3 quality control methods: Rule-based filtering, LLM-based filtering, and Human evaluation. The implementation details and specific results of the quality control are presented in Appendix E.2.

**Benchmark Construction** Leveraging the multi-dimensional assessing methodology for scientific logicality, we introduce PHYSLOGIC – the first comprehensive benchmark for logical reasoning in physics. Specifically, we selected a total of 864 papers from nine distinct physics subfields. For each subfield, we curated a balanced set of 96 questions spanning four difficulty levels (High School, Undergraduate, Master's, and PhD) and four problem types. To ensure a comprehensive and balanced evaluation, each difficulty-type combination comprises 6 distinct problems. The innovative aspects of our benchmark, compared to existing work, are summarized in Table 2.

**SFT Data Construction** Beyond the 864 instances reserved for our benchmark, we randomly sampled 80k entries to generate data for SFT. Following the sampling methods in Section 2.2, we constructed two instruction-tuning datasets with high logicality: 80k samples for Reasoning Style Transfer (**RST**) and 40k for Distillation with Logic Supervision (**Logic-Distill**). In addition, an 80k-sample baseline dataset was created using the direct reasoning outputs of R1 (**Direct-Distill**).

**Dataset Statistics** We conducted a statistical analysis of the content and distribution of the constructed dataset. Table 1 summarizes the statistics for the 80k training dataset, categorized by four tasks. It details the proportions, the average token counts for questions, reasonings and answers, the average number and length of logical nexuses. Additional statistics and visualizations for the dataset can be found in Appendix E.1.

---

[4]https://api-docs.deepseek.com

# 4. Experiments

In this section, we empirically evaluate our proposed methodology and aim to answer three key questions. We examine Q1 via two empirical studies, Q2 via in-domain experiments on our proposed PHYSLOGIC benchmark, and Q3 via out-of-domain experiments on public physics QA benchmarks.

- **Q1:** *Do our proposed metrics genuinely capture the logicality of reasoning?*

- **Q2:** *Can our proposed logicality-based SFT data sampling method enhance the scientific logicality of LLMs in physics reasoning?*

- **Q3:** *Does the improved scientific logicality really contribute to better task performance?*

## 4.1. Training and Evaluation Setup

**Model Training** From the constructed dataset, we sample three SFT subsets: (1) Direct-Distill (80k), (2) Reasoning Style Transfer (RST, 80k), and (3) Logic-Distill (40k). The backbone LLMs include 1) a reasoning LLM: DeepSeek-R1-Distill-Qwen-7B (Hereinafter referred to as "R1-7B") (Guo et al., 2025); 2) a chat LLM: Qwen2.5-7B-Instruct (Hereinafter referred to as "Qwen2.5-7B") (Yang et al., 2025); and 3) a base LLM: Llama-3.1-8B (Grattafiori et al., 2024).

During the training period, we employed the efficient `LlamaFactory`[5] framework to perform full-parameter fine-tuning on the model. To ensure training stability and efficiency, we meticulously configured a series of hyperparameters. Specifically, the learning rate was set to $5.0 \times 10^{-6}$, paired with a cosine learning rate scheduler for dynamic adjustments, and a warmup ratio of $0.03$. Given computational resource constraints, we set the per-device train batch size to $1$ and used $2$ gradient accumulation steps, achieving an effective batch size of $2$. Additionally, to handle long text sequences, the model's maximum sequence length (cutoff length) was extended to $32768$.

For optimization, we adopted several advanced techniques to enhance training efficiency and reduce memory consumption. BF16 mixed-precision was enabled throughout the training process, and the `DeepSpeed ZeRO Stage 3`[6] optimization strategy was integrated. Furthermore, we applied the `FlashAttention-2`[7] mechanism to accelerate the computation of the attention module and enabled gradient checkpointing to further conserve memory.

To ensure the reproducibility of our experiments, the global random seed was fixed to $42$. The entire training process

was conducted for $2$ epochs. The training for each model was conducted on 8 NVIDIA H100 Tensor Core GPUs.

**Baselines** Besides the Direct-Distill baseline, we also benchmark against four public physics-QA datasets: NaturalReasoning (Yuan et al., 2025), MegaScience (Fan et al., 2025), SCP-116k (Lu et al., 2025), and Sci-Instruct (Zhang et al., 2024a). For a fair comparison, we only use the physics-related subsets from these datasets and sample an equal amount of data (80k) for each training set. Details of the baseline datasets are provided in Appendix F.1

**Evaluation Metrics** In-domain, we evaluate on PHYSLOGIC using Logical Fidelity ($\mathcal{F}$), Causal Connection ($\mathcal{O}$), Inferential Progress ($\mathcal{P}$), and final-answer accuracy on multiple-choice and numerical problems (Acc). To avoid methodological circularity, since the "Logic-Distill" data are sampled using PHYSLOGIC's metrics, we report in-domain results by comparing only against "RST" to ensure objective evaluation. Out-of-domain, we report final accuracy on three public benchmarks with distinct formats: (1) multiple choice, physics subset of GPQA-Diamond (GPQA) (Rein et al., 2024); (2) numerical calculation, physics subset of SciBench (SciBench) (Wang et al., 2024); and (3) reasoning, PhysReason (PhysR.) (Zhang et al., 2025). All scores are average percentages over three independent runs. For logicality evaluation, the sentence encoder is `all-MiniLM-L6-v2`[8]. The prompts used for the inference phase and for the judge LLMs across all four benchmarks are provided in Appendix I.4. More details are in Appendix F.2.

## 4.2. Validity of Logicality Metrics

To answer **Q1:** *"Do our proposed metrics genuinely capture the logicality of reasoning?"*, in this section, we design two empirical experiments to validate the validity of the three proposed metrics $\mathcal{F}$, $\mathcal{O}$ and $\mathcal{P}$ before the main experiment.

**Study 1: Consistency with third-party evaluation indicators.** We randomly sample 200 instances from our benchmark. A human physics expert and ChatGPT-5 each assign 1–10 logicality scores to the reasoning processes produced by R1-7B. The scoring rubric is reported in Appendix H. We then compute Pearson and Spearman correlation coefficients (Pearson, 1895; Spearman, 1904) between each component metric ($\mathcal{F}$, $\mathcal{O}$, $\mathcal{P}$) as well as their averaged score (Overall), and the human/LLM judger ratings. As shown in Table 3, all three metrics exhibit positive and significant correlations with both third-party indicators (all $p < 0.001$), suggesting that **our automatic logicality metrics align well with human and LLM judgments.**

---

*Table 3.* Consistency between third-party ratings and our logicality metrics, measured by Pearson/Spearman correlations (All correlations are statistically significant, $p < 0.001$).

| | Pearson | | | | Spearman | | | |
|---|---|---|---|---|---|---|---|---|
| Rater | $\mathcal{F}$ | $\mathcal{O}$ | $\mathcal{P}$ | Avg. | $\mathcal{F}$ | $\mathcal{O}$ | $\mathcal{P}$ | Avg. |
| Human Expert | 0.8021 | 0.7114 | 0.6948 | 0.7453 | 0.8157 | 0.7249 | 0.7092 | 0.7798 |
| LLM Judge | 0.8263 | 0.7436 | 0.7471 | 0.7860 | 0.8404 | 0.7582 | 0.7717 | 0.8303 |

*Table 4.* Means and medians of logicality metrics on correct/incorrect samples.

| | $\mathcal{F}$ | | $\mathcal{O}$ | | $\mathcal{P}$ | | Avg. | |
|---|---|---|---|---|---|---|---|---|
| | mean | median | mean | median | mean | median | mean | median |
| Correct | **52.3** | **53.6** | **73.1** | **76.9** | **6.37** | **5.12** | **43.9** | **43.7** |
| Incorrect | 46.0 | 50.0 | 67.5 | 72.1 | 5.55 | 4.78 | 39.7 | 39.3 |

*Table 5.* In-domain experimental results on our proposed PHYSLOGIC benchmark

| Backbones | Llama-3.1-8B-base | | | | | Qwen2.5-7B-Instruct | | | | | DeepSeek-R1-Distill-Qwen-7B | | | | |
|---|---|---|---|---|---|---|---|---|---|---|---|---|---|---|---|
| SFT Data | $\mathcal{F}$ | $\mathcal{O}$ | $\mathcal{P}$ | Avg. | Acc | $\mathcal{F}$ | $\mathcal{O}$ | $\mathcal{P}$ | Avg. | Acc | $\mathcal{F}$ | $\mathcal{O}$ | $\mathcal{P}$ | Avg. | Acc |
| **NaturalReasoning** | 53.56 | 70.49 | 2.96 | 42.34 | 23.61 | 52.43 | 70.76 | 2.97 | 42.05 | 35.88 | 48.08 | 59.85 | 7.63 | 38.52 | 35.65 |
| **MegaScience** | 53.71 | 68.10 | 5.25 | 42.35 | 31.02 | 54.63 | 69.25 | **5.49** | 43.12 | 39.81 | 39.96 | 71.08 | 3.93 | 38.32 | 44.44 |
| **Sci-Instruct** | 50.76 | 61.59 | 4.47 | 38.94 | 13.89 | 45.68 | 63.11 | 4.35 | 37.71 | 15.74 | 51.03 | 61.52 | 7.04 | 39.86 | 32.87 |
| **SCP-116k** | 46.60 | 63.57 | 4.39 | 38.19 | 25.00 | 47.09 | 63.91 | 4.28 | 38.43 | 37.27 | 53.66 | 67.48 | 6.89 | 42.68 | 46.30 |
| **Ours (RST)** | **58.70** | **73.69** | **5.43** | **45.94** | **44.67** | **58.89** | **71.16** | 5.12 | **45.06** | **42.82** | **56.68** | **73.54** | **7.71** | **45.98** | **47.45** |

For each matrix, the best results highlighted in **bold** and the second-best results underlined.

*Table 6.* Comparative results on public physics benchmarks between our constructed data and baselines

| Backbone | Data | Llama-3.1-8B | | | | Qwen2.5-7B-Instruct | | | | DeepSeek-R1-Distill-Qwen-7B | | | |
|---|---|---|---|---|---|---|---|---|---|---|---|---|---|
| SFT Dataset | Scale | GPQA | SciBench | PhysR. | Avg. | GPQA | SciBench | PhysR. | Avg. | GPQA | SciBench | PhysR. | Avg. |
| **NaturalReasoning** | 80k | 7.36 | 17.09 | 14.48 | 12.98 | 23.26 | 30.05 | 17.81 | 23.71 | 36.82 | 26.94 | 20.33 | 28.03 |
| **MegaScience** | 80k | 27.02 | 15.02 | 16.33 | 19.46 | 36.82 | 32.12 | 19.22 | 29.39 | 53.49 | 50.94 | 26.80 | 43.74 |
| **Sci-Instruct** | 80k | 19.77 | 7.25 | 13.72 | 13.58 | 30.62 | 9.84 | 17.19 | 19.22 | 34.50 | 7.14 | 20.15 | 20.60 |
| **SCP-116k** | 80k | 43.02 | 32.64 | 29.57 | 35.08 | 41.47 | 43.52 | 19.17 | 34.72 | 56.59 | 52.33 | 33.09 | 47.34 |
| **Ours (Direct-Distill)** | 80k | **46.90** | **37.82** | 29.21 | **37.98** | 37.98 | 48.70 | 22.18 | 36.29 | 51.55 | 50.77 | 30.50 | 44.27 |
| **Ours (Logic-Distill)** | **40k** | 39.53 | 35.75 | **30.13** | 35.14 | 43.02 | **49.22** | **42.88** | **45.04** | 53.49 | 53.71 | **53.05** | **53.42** |
| **Ours (RST)** | 80k | 40.70 | 27.46 | 24.77 | 30.98 | **47.67** | 38.86 | 37.71 | 41.41 | **60.46** | **55.95** | 40.36 | 52.26 |

* For each backbone, the best results are highlighted in **bold** and the second-best results are underlined.

**Study 2: Relation between logicality and task performance.** We further examine whether higher logicality scores are associated with better task performance. For Qwen2.5-7B, R1-7B, and GPT-5, we compute $\mathcal{F}$, $\mathcal{O}$, and $\mathcal{P}$ for correctly and incorrectly answered samples, as well as their average (Overall). Table 4 reports the scores of the two groups. Across all three dimensions and Avg., correct samples obtain significantly higher scores than incorrect ones ($p < 0.001$), showing that **higher logicality is closely associated with better reasoning performance**.

### 4.3. In-Domain Experiment

In this section, we systematically evaluate the scientific logicality of existing LLMs on physics reasoning and examine whether our supervised fine-tuning (SFT) data can improve this capability. We compare four baseline datasets with our RST-sampled dataset by fine-tuning three backbones on the same 80k training examples and evaluating scientific logicality. As shown in Table 5, SFT on RST-sampled trajectories consistently yields the highest scientific logicality across all three backbones, outperforming the second-best baseline by **3.59%**, **1.94%**, and **3.30%** in average logicality, respectively; it also improves final answer accuracy by **13.65%**, **3.01%**, and **1.15%** over the second-best results. We further evaluate a broader set of open- and closed-source LLMs on our PHYSLOGIC benchmark. Figures 5, 6, 7 and 8 in Appendix C provide a direct comparison of scientific logicality across models, showing that fine-tuning a 7B model on only our 80k training set surpasses comparable 14B and even 32B models overall, and achieves the highest average logicality among all closed-source LLMs. Together, these results provide an affirmative answer to **Q2**: **our RST-based SFT data effectively enhances LLM scientific logicality in physics reasoning**.

*Table 7.* Ablation study on different backbones and settings.

| Backbone | Llama-3.1-8B | | Qwen-2.5-7B-Instruct | | DeepSeek-R1-Distill-Qwen-7B | |
|---|---|---|---|---|---|---|
| Setting | Logicality | Answer | Logicality | Answer | Logicality | Answer |
| Logic-Distill | **45.50** | **36.54** | **42.78** | **44.02** | **44.14** | **49.73** |
| w/o $\mathcal{F}$ | 43.90 (-1.60) | 33.85 (-2.69) | 40.03 (-2.75) | 40.59 (-3.43) | 41.58 (-2.56) | 45.08 (-4.65) |
| w/o $\mathcal{O}$ | 44.05 (-1.85) | 31.31 (-5.23) | 38.35 (-4.43) | 38.25 (-5.77) | 41.38 (-2.76) | 38.68 (-11.05) |
| w/o $\mathcal{P}$ | 44.06 (-1.44) | 33.72 (-2.82) | 40.77 (-2.01) | 38.72 (-5.30) | 41.69 (-2.45) | 45.18 (-4.55) |
| Random | 43.67 (-1.83) | 31.93 (-4.61) | 41.73 (-1.03) | 28.77 (-15.25) | 40.24 (-3.90) | 41.78 (-7.95) |

\* The best results highlighted in **bold**, in ablation results, the parenthesized deltas following each metric denote the change with respect to "Logic-Distill".

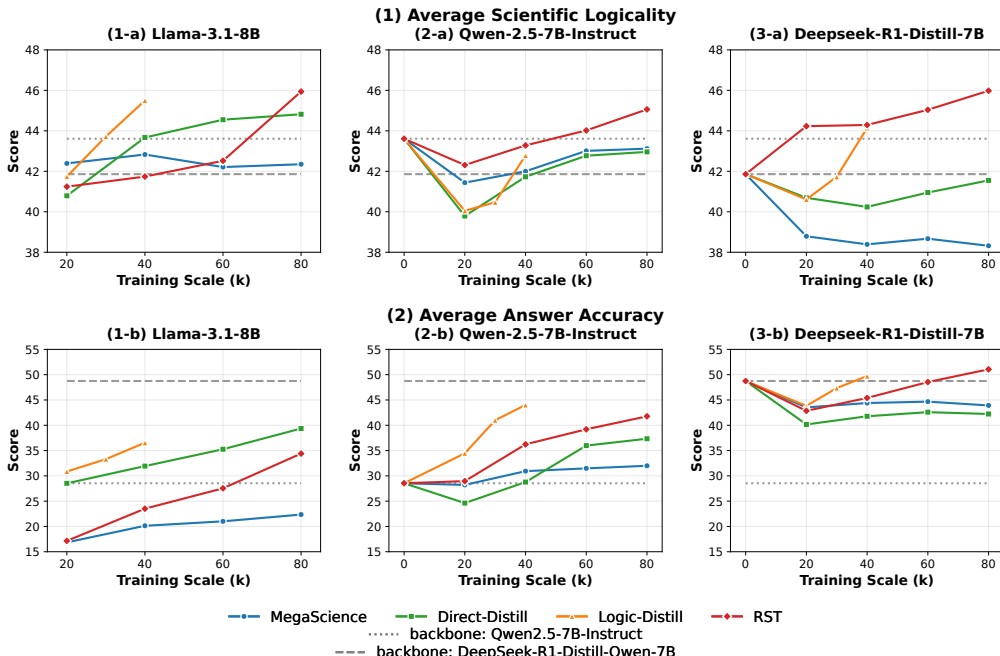

*Figure 4.* Scaling law curves for scientific logicality and task performance of models trained on four SFT datasets at varying data scales.

### 4.4. Out-of-Domain Experiment

Table 6 reports the evaluation results on three public benchmarks. When further trained on Qwen2.5-7B and R1-7B, our proposed "Logic-Distill" and "RST" methods outperform the other baselines. Notably, "Logic-Distill", which incorporates finer-grained computational steps, performs best: despite using only *half of the training data* compared to all the other baselines, it outperforms the strongest baseline by **8.75%** and **6.08%** on Qwen2.5-7B and R1-7B, respectively. "RST" ranks second, exceeding the best baseline by **5.12%** and **4.92%** on the two backbones. On the Llama-3.1-8B-base model, "Direct-Distill" outperforms "Logic-Distill", which we attribute to more pronounced scaling-law effects on base models, as the former is trained with a larger dataset of 80k reasoning instances. However, a comparison with an equivalent amount of training data (refer to the next section "Scaling Laws") demonstrates that "Logic-Distill" is better and more efficient. These findings provide a conclusive answer to **Q3: training with higher-logicality reasoning data also positively impacts the final performance of LLMs on various public physics QA tasks**.

### 4.5. Ablation Study

We ablate our three logicality dimensions ($\mathcal{F}$, $\mathcal{O}$, $\mathcal{P}$) by removing each one individually from the "Logical-Distill" sampling process, re-weighting the other two to 0.5. As shown in Table 7, removing any single dimension significantly degrades both scientific logicality and task performance. The ablation of Causal Connection ($\mathcal{O}$), which assesses reasoning order, causes the most substantial performance drop, due to its role in filtering out hallucinations.

### 4.6. Scaling Law

We study SFT effects across backbones by varying data size for "MegaScience", "Direct-Distill", "Logic-Distill", and "RST" and plotting scaling-law curves of mean logicality and accuracy (Figure 4). On Qwen and DeepSeek, perfor-

mance typically dips before rising, likely due to a reasoning paradigm mismatch between SFT traces and the native pre-training data that hurts small-data SFT. The growth trends for our "Logic-Distill" and "RST" are the most pronounced. Moreover, the comparison between "Logic-Distill" and "Direct-Distill" at equivalent data volumes clearly demonstrates that for SFT with scientific reasoning data, enhancing scientific logicality is a more effective strategy than simply increasing the data scale.

### 4.7. Supplementary Experiments

In addition to the experiments described above, we conducted more extensive analyses and studies, including scalability to larger backbones (Appendix D.1), OOD performance on mathematical benchmarks (Appendix D.2), different matching strategies (Appendix D.3), the logicality score of the training set (Appendix D.4), and the ablation on sampling percentiles in "Logic-Distill" (Appendix D.5). Furthermore, to intuitively illustrate the evaluative role of our three logicality metrics, we provide case studies in Appendix K.

## 5. Related Work

**Dataset and benchmarks for LLM physics reasoning** Supervision typically derives from corpus extraction or LLM-based synthesis. NATURALREASONING and SCP-116K extract QA from research corpora and textbooks, with explicit step traces in the latter (Yuan et al., 2025; Lu et al., 2025); SCI-INSTRUCT synthesizes and self-revises data (Zhang et al., 2024a); MEGASCIENCE aggregates public datasets with difficulty-aware filtering (Fan et al., 2025). Evaluation includes cross-disciplinary suites: GPQA (MCQ) and SCIBENCH (open-ended numerical problems) (Rein et al., 2024; Wang et al., 2024); and physics-specific sets: UGPHYSICS (undergraduate exercises with rule-based scoring) and PHYSREASON (competition problems with step-level verification) (Xu et al., 2025; Zhang et al., 2025).

**Process-oriented evaluation of LLM reasoning** These methods assess step validity rather than only final answers. REVEAL labels relevance, attribution, and logical correctness (Jacovi et al., 2024); PROCESSBENCH and PRM-BENCH evaluate step-error detection and PRM robustness (Zheng et al., 2025a; Song et al., 2025); PRISM-PHYSICS uses a topological structure to verify the problem-solving process (Zhao et al., 2025), but it only touches upon one aspect of logicality; VERIFYBENCH extends verification to physics, chemistry, and biology (Li et al., 2026).

## 6. Conclusion

This work pioneers a systematic study of scientific logicality in LLM reasoning. We introduce assessment criteria

to quantify this logicality and propose two SFT data sampling strategies to effectively improve it. Taking physics as an exemplar discipline, we construct a dedicated dataset and benchmark to practise our methodology. Empirical studies involving third-party verification demonstrate the effectiveness of the proposed logical metrics. Comprehensive experiments verify the effectiveness of our proposed methodology for both scientific logicality and task performances in physics.

## Acknowledgments

This work is supported in part by the National Natural Science Foundation of China under Grants #72293575, #72225011 and #72434005, and the Joint Research Project on the Integration of Culture, Science and Technology between Chinese Academy of Sciences and Hunan Province (#2024JK4003).

## Impact Statement

This work studies scientific logicality in LLM reasoning and proposes assessment criteria and logic-aware data sampling methods for physics reasoning. By shifting evaluation and training from final-answer accuracy alone toward logical fidelity, causal connection, and inferential progress, our benchmark and datasets may support more reliable process-level evaluation of scientific reasoning models and contribute to AI-assisted scientific education, research assistance, and model development. This process-level perspective also helps identify where a model's scientific reasoning fails, rather than only whether its final answer is wrong.

However, process-level logicality does not guarantee factual correctness, scientific validity, or safe deployment. Our metrics assess reasoning traces by alignment with extracted logical nexuses, causal ordering, and inferential progression, and should be viewed as diagnostic tools rather than certificates of correctness. Models may still produce plausible but incorrect derivations, especially beyond the physics settings studied here. We therefore recommend combining our metrics with final-answer evaluation, expert inspection, and domain-specific robustness tests, while treating such systems as decision-support tools rather than expert substitutes.

Finally, our dataset and benchmark are constructed from scholarly articles in public repositories, including arXiv and established physics journals. To reduce ethical and privacy risks, we remove metadata such as author names and affiliations, constrain generation to central scientific problems, and apply rule-based and LLM-based filters together with human evaluation for quality control. In our public release, we will provide only synthesized question-answer pairs and logical nexuses, without information intended to identify specific source papers.

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

# Appendices

## Contents

# A. Reproducibility Statement

To ensure the reproducibility of the research results in this work, we provide the following details:

- **Training Details:** Section 4.1 provides the detailed parameters and hardware specifications for the model training process.

- **Evaluation Details:** Appendix F presents details of the four public baseline datasets, the specific implementation methods for the benchmarking process, and the model deployment details, respectively.

- **Parameter Sensitivity Analysis:** Appendix G details the parameters involved in our proposed method, along with a sensitivity analysis of these parameters.

- **Complete Prompts:** Appendix I provides all the prompts used to query the LLMs throughout the entire workflow of this work.

- **Open-Source Code:** PHYSLOGIC benchmark and the evaluation codes are available at https://github.com/ScienceOne-AI/PhysLogic.

# B. Complete Experimental Results

Due to space limitations, we cannot present all the detailed results of the scaling law experiments and ablation study in the main text. This chapter, however, includes Tables 8, 9 and 10, which report the complete results of the scaling law experiments using three different backbone LLMs, and Table 11, which reports the complete results of the ablation study.

# C. Performances of Various LLMs on Our PHYSLOGIC Benchmark

We tested a total of 25 types of open-source (in blue)/closed-source (in purple) LLMs and LLMs after sft using the data we constructed (in yellow), and conducted comparative evaluations on the PhysLogic dataset we constructed. Figures 5, 6 and 7 reports the three logicality scores. Figure 8 reports the final accuracy.

# D. Supplementary Experiments

## D.1. Scalability to larger backbones

To further examine whether the effectiveness of our method scales to larger language models, we conduct additional experiments using Qwen-2.5-14B-Instruct as the backbone. We compare Logic-Distill (LD) and RST with two baselines, MegaScience and Direct-Distill (DD), and evaluate them on PhysLogic, GPQA, and SciBench. The results are summarized in Table 12.

As shown in Table 12, both LD and RST consistently improve the average logicality score over the baselines. In terms of final-answer accuracy, LD achieves the best average performance, improving from 38.41 with MegaScience and 39.76 with DD to 45.28. RST obtains a comparable average accuracy of 45.26, while achieving the highest average logicality score. These results indicate that the benefits of our logic-guided training strategy are not limited to smaller backbones, but can also transfer effectively to larger-scale LLMs.

## D.2. Out-of-domain evaluation on math benchmarks

To examine whether our physics-specific training corpus can generalize beyond the physics domain, we further conduct out-of-domain experiments on mathematical reasoning benchmarks. Using Qwen-2.5-7B-Instruct as the backbone, we compare Direct-Distill, Logic-Distill, and RST against MegaScience, a scientific reasoning distillation baseline. We evaluate all methods on four math benchmarks, including MATH-500[9], GSM8K[10], AIME2025[11], and AMC[12]. All scores are averaged over 8 independent runs. The results are summarized in Table 13.

---

[9]https://huggingface.co/datasets/HuggingFaceH4/MATH-500

[10]https://huggingface.co/datasets/openai/gsm8k

[11]https://huggingface.co/datasets/math-ai/aime25

[12]https://huggingface.co/datasets/math-ai/amc23

*Table 8.* Complete results of scaling law experiment based on Llama-3.1-8B.

| Dataset | Scale | Public Benchmarks | | | | PhysLogic | | | | |
| | | GPQA-D$_{(Phy.)}$ | SciBench$_{(Phy.)}$ | PhysReason | Average | $\mathcal{F}$ | $\mathcal{O}$ | $\mathcal{P}$ | Average Logicality | Answer Score |
|---|---|---|---|---|---|---|---|---|---|---|
| MegaScience | 20k | 18.21 | 11.92 | 11.59 | 13.91 | 53.68 | 68.46 | 5.04 | 42.39 | 25.69 |
| | 40k | 21.32 | 13.99 | 14.66 | 16.66 | 54.03 | 67.38 | 7.07 | 42.83 | 30.56 |
| | 60k | 25.97 | 13.47 | 15.90 | 18.45 | 53.63 | 68.09 | 4.92 | 42.21 | 28.70 |
| | 80k | 27.02 | 15.02 | 16.33 | 19.46 | 53.71 | 68.10 | 5.25 | 42.35 | 31.02 |
| Ours (Direct-Distill) | 20k | 33.72 | 24.87 | 23.84 | 27.48 | 49.09 | 69.28 | 4.01 | 40.79 | 31.64 |
| | 40k | 39.15 | 32.30 | 16.70 | 29.38 | 56.00 | 70.31 | 4.70 | 43.67 | 39.58 |
| | 60k | 39.92 | 37.30 | 24.71 | 33.98 | 56.65 | 72.05 | 4.96 | 44.55 | 39.12 |
| | 80k | 46.90 | 37.82 | 29.21 | 37.98 | 56.97 | 72.48 | 5.00 | 44.82 | 43.52 |
| Ours (Logic-Distill) | 10k | 32.17 | 23.31 | 23.23 | 26.24 | 47.13 | 67.28 | 3.97 | 39.46 | 24.54 |
| | 20k | 36.05 | 27.46 | 27.73 | 30.41 | 50.69 | 70.20 | 4.32 | 41.74 | 32.18 |
| | 30k | 37.21 | 30.57 | 28.84 | 32.21 | 54.11 | 72.52 | 4.54 | 43.72 | 36.57 |
| | 40k | 39.53 | 35.75 | 30.13 | 35.14 | 57.38 | 74.28 | 4.84 | 45.50 | 40.74 |
| Ours (RST) | 20k | 18.22 | 10.88 | 15.90 | 15.00 | 52.18 | 66.69 | 4.85 | 41.24 | 23.61 |
| | 40k | 25.97 | 19.67 | 20.15 | 21.93 | 53.90 | 67.05 | 4.28 | 41.74 | 28.24 |
| | 60k | 31.01 | 22.28 | 22.37 | 25.22 | 54.04 | 68.98 | 4.55 | 42.52 | 34.49 |
| | 80k | 40.70 | 27.46 | 24.77 | 30.98 | 58.70 | 73.69 | 5.43 | 45.94 | 44.67 |

*Table 9.* Complete results of scaling law experiment based on Qwen-2.5-7B-Instruct.

| Dataset | Scale | Public Benchmarks | | | | PhysLogic | | | | |
| | | GPQA-D$_{(Phy.)}$ | SciBench$_{(Phy.)}$ | PhysReason | Average | $\mathcal{F}$ | $\mathcal{O}$ | $\mathcal{P}$ | Average Logicality | Answer Score |
|---|---|---|---|---|---|---|---|---|---|---|
| backbone | 0 | 25.97 | 37.30 | 16.64 | 26.64 | 57.65 | 66.73 | 6.44 | 43.61 | 34.26 |
| MegaScience | 20k | 27.91 | 25.39 | 23.66 | 25.65 | 51.49 | 66.49 | 6.33 | 41.44 | 35.88 |
| | 40k | 33.72 | 27.98 | 24.96 | 28.89 | 52.28 | 67.10 | 6.63 | 42.00 | 37.04 |
| | 60k | 34.11 | 29.02 | 24.58 | 29.24 | 55.37 | 68.06 | 5.59 | 43.01 | 38.19 |
| | 80k | 36.82 | 32.12 | 19.22 | 29.39 | 54.63 | 69.25 | 5.49 | 43.12 | 39.81 |
| Ours (Direct-Distill) | 20k | 24.42 | 19.17 | 17.92 | 20.50 | 47.93 | 67.04 | 4.39 | 39.79 | 36.96 |
| | 40k | 27.51 | 30.05 | 18.55 | 25.37 | 51.64 | 69.30 | 4.24 | 41.73 | 38.97 |
| | 60k | 37.21 | 45.60 | 20.83 | 34.55 | 53.09 | 70.85 | 4.36 | 42.77 | 40.28 |
| | 80k | 37.98 | 48.70 | 22.18 | 36.29 | 53.92 | 70.35 | 4.61 | 42.96 | 40.51 |
| Ours (Logic-Distill) | 10k | 18.60 | 30.22 | 33.39 | 27.40 | 45.29 | 66.98 | 4.31 | 38.86 | 35.42 |
| | 20k | 29.07 | 34.89 | 36.30 | 33.42 | 48.07 | 67.87 | 4.22 | 40.05 | 37.58 |
| | 30k | 40.71 | 42.49 | 42.14 | 41.78 | 48.23 | 68.66 | 4.52 | 40.47 | 38.66 |
| | 40k | 43.02 | 49.22 | 42.88 | 45.04 | 53.98 | 69.39 | 4.97 | 42.78 | 40.97 |
| Ours (RST) | 20k | 26.74 | 26.42 | 27.73 | 26.96 | 53.68 | 68.39 | 4.86 | 42.31 | 34.95 |
| | 40k | 36.05 | 36.79 | 34.57 | 35.80 | 54.96 | 69.71 | 5.18 | 43.28 | 37.50 |
| | 60k | 41.86 | 38.34 | 35.61 | 38.60 | 55.48 | 71.59 | 4.98 | 44.02 | 40.97 |
| | 80k | 47.67 | 38.86 | 37.71 | 41.41 | 58.89 | 71.16 | 5.12 | 45.06 | 42.82 |

*Table 10.* Complete results of scaling law experiment based on DeepSeek-R1-Distill-Qwen-7B.

| Dataset | Scale | Public Benchmarks | | | | PhysLogic | | | | |
| | | GPQA-D$_{(Phy.)}$ | SciBench$_{(Phy.)}$ | PhysReason | Average | $\mathcal{F}$ | $\mathcal{O}$ | $\mathcal{P}$ | Average Logicality | Answer Score |
|---|---|---|---|---|---|---|---|---|---|---|
| backbone | 0 | 66.28 | 60.1 | 28.1 | 51.49 | 47.91 | 66.78 | 10.9 | 41.86 | 40.51 |
| MegaScience | 20k | 45.35 | 51.3 | 32.08 | 42.91 | 42.3 | 69.49 | 4.57 | 38.79 | 45.37 |
| | 40k | 52.33 | 50.78 | 31.88 | 45.00 | 40.88 | 69.85 | 4.45 | 38.39 | 42.59 |
| | 60k | 52.71 | 50.78 | 30.75 | 44.75 | 41.42 | 70.27 | 4.31 | 38.67 | 44.44 |
| | 80k | 53.49 | 50.94 | 26.8 | 43.74 | 39.96 | 71.08 | 3.93 | 38.32 | 44.44 |
| Ours (Direct-Distill) | 20k | 44.19 | 48.7 | 37.03 | 43.31 | 49.52 | 68 | 4.54 | 40.69 | 30.71 |
| | 40k | 48.45 | 51.3 | 32.16 | 43.97 | 50.06 | 66.1 | 4.56 | 40.24 | 35.19 |
| | 60k | 51.16 | 51.3 | 32.47 | 44.98 | 51.11 | 67.06 | 4.68 | 40.95 | 35.42 |
| | 80k | 51.55 | 50.77 | 30.5 | 44.27 | 51.31 | 68.82 | 4.53 | 41.55 | 36.11 |
| Ours (Logic-Distill) | 10k | 43.02 | 43.52 | 45.66 | 44.07 | 48.93 | 66.51 | 4.43 | 39.96 | 33.1 |
| | 20k | 44.19 | 50.25 | 48.73 | 47.72 | 50.28 | 67.08 | 4.47 | 40.61 | 32.18 |
| | 30k | 48.84 | 54.4 | 51.2 | 51.48 | 52.2 | 68.23 | 4.77 | 41.73 | 34.95 |
| | 40k | 53.49 | 53.71 | 53.05 | 53.42 | 55.33 | 71.9 | 5.19 | 44.14 | 38.66 |
| Ours (RST) | 20k | 44.96 | 51.81 | 32.97 | 43.25 | 54.38 | 68.4 | 9.91 | 44.23 | 41.67 |
| | 40k | 50 | 50.78 | 35.67 | 45.48 | 54.83 | 69.08 | 8.97 | 44.29 | 45.14 |
| | 60k | 57.36 | 55.44 | 34.38 | 49.06 | 56.58 | 71.4 | 7.14 | 45.04 | 46.99 |
| | 80k | 60.46 | 55.95 | 40.36 | 52.26 | 56.68 | 73.54 | 7.71 | 45.98 | 47.45 |

*Table 11.* Complete results of ablation study.

| Backbone | Setting | Public Benchmarks | | | | PhysLogic | | | | |
|---|---|---|---|---|---|---|---|---|---|---|
| | | GPQA-D$_{(Phy.)}$ | SciBench$_{(Phy.)}$ | PhysReason | Average | $\mathcal{F}$ | $\mathcal{O}$ | $\mathcal{P}$ | Average Logicality | Answer Score |
| Llama -3.1-8B | Logic-Distill | 39.53 | 35.75 | 30.13 | 35.14 | 57.38 | 74.28 | 4.84 | 45.50 | 40.74 |
| | w/o F | 46.51 | 30.05 | 20.64 | 32.40 | 54.66 | 72.37 | 4.94 | 43.99 | 38.19 |
| | w/o O | 38.37 | 28.5 | 20.39 | 29.09 | 54.8 | 72.63 | 4.73 | 44.05 | 37.96 |
| | w/o P | 38.37 | 34.71 | 24.77 | 32.62 | 55.1 | 72.34 | 4.75 | 44.06 | 37.04 |
| | random | 39.15 | 32.3 | 16.7 | 29.38 | 56 | 70.31 | 4.7 | 43.67 | 39.58 |
| Qwen2.5-7B -Instruct | Logic-Distill | 43.02 | 49.22 | 42.88 | 45.04 | 53.98 | 69.39 | 4.97 | 42.78 | 40.97 |
| | w/o F | 41.86 | 45.08 | 38.15 | 41.70 | 48.64 | 67.41 | 4.04 | 40.03 | 37.27 |
| | w/o O | 39.53 | 42.66 | 36.1 | 39.43 | 45.12 | 65.16 | 4.78 | 38.35 | 34.72 |
| | w/o P | 41.09 | 46.98 | 29.08 | 39.05 | 48.96 | 69.22 | 4.13 | 40.77 | 37.73 |
| | random | 27.51 | 30.05 | 18.55 | 25.37 | 51.64 | 69.3 | 4.24 | 41.73 | 38.97 |
| DeepSeek-R1 -Distill -Qwen-7B | Logic-Distill | 53.49 | 53.71 | 53.05 | 53.42 | 55.33 | 71.9 | 5.19 | 44.14 | 38.66 |
| | w/o F | 53.1 | 46.62 | 46.58 | 48.77 | 52.76 | 67.39 | 4.6 | 41.58 | 34.03 |
| | w/o O | 45.35 | 49.22 | 26.1 | 40.22 | 51.54 | 67.72 | 4.89 | 41.38 | 34.03 |
| | w/o P | 51.94 | 51.3 | 40.42 | 47.89 | 52.16 | 68.04 | 4.87 | 41.69 | 37.04 |
| | random | 48.45 | 51.3 | 32.16 | 43.97 | 50.06 | 66.1 | 4.56 | 40.24 | 35.19 |

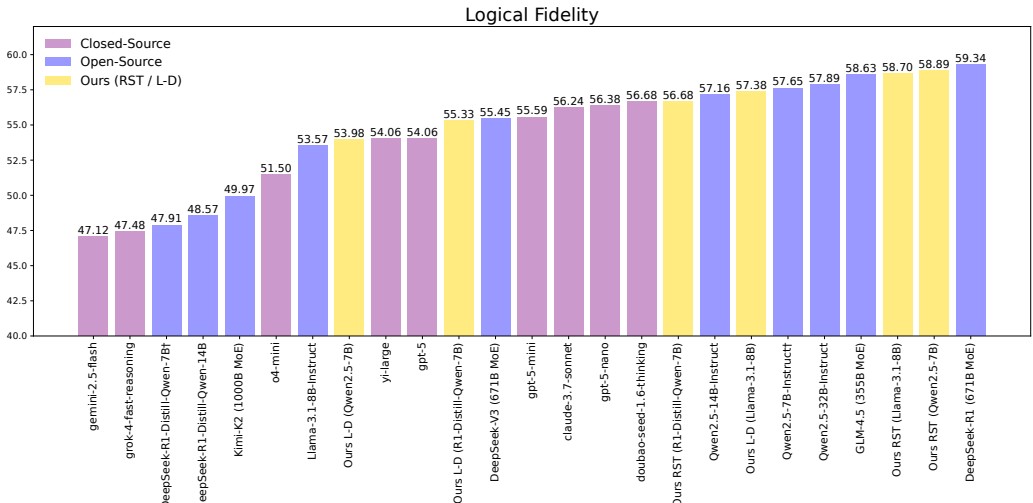

*Figure 5.* Visualization of logical fidelity score

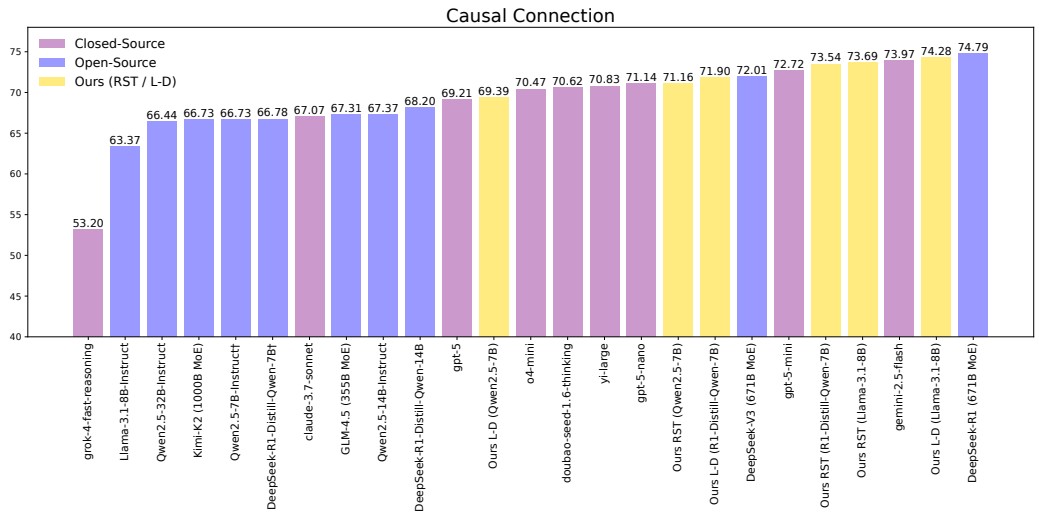

*Figure 6.* Visualization of causal connection score

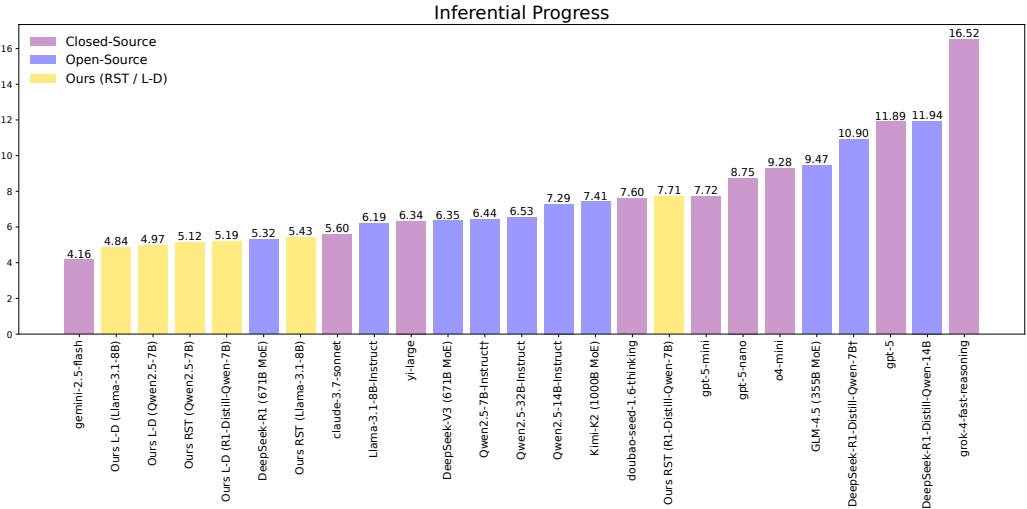

*Figure 7.* Visualization of inferential progress score

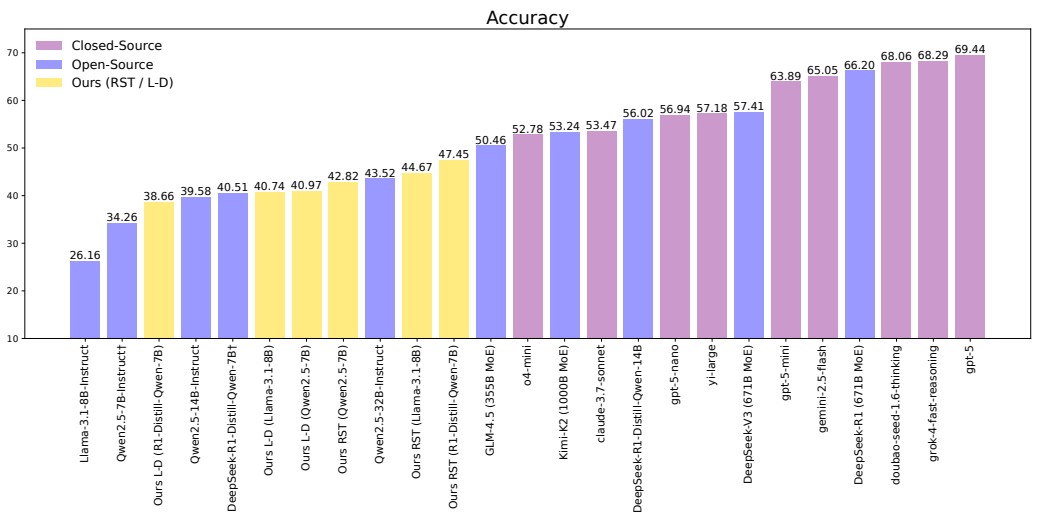

*Figure 8.* Visualization of final answer accuracy

Although our constructed training set is purely physics-oriented, it still yields non-trivial improvements on mathematical reasoning tasks. In particular, Logic-Distill achieves the best performance on all four benchmarks, improving the average score from 51.56 with MegaScience and 50.76 with Direct-Distill to 54.54. RST also achieves a higher average score than both baselines. These results provide further evidence that our logic-guided training strategy exhibits meaningful cross-domain generalization beyond physics.

### D.3. Sensitivity to the matching strategy for logical fidelity

In our main experiments, we adopt a greedy matching strategy to compute logical fidelity $\mathcal{F}$. This choice is primarily motivated by computational efficiency, since the metric must be evaluated many times over long reasoning trajectories, across multiple datasets and models. More complex global alignment methods would substantially increase the runtime of the evaluation pipeline.

To assess whether our conclusions are sensitive to this design choice, we additionally implement a global alignment method based on dynamic programming and recompute the logical fidelity scores for all evaluated models on the same set of reasoning processes. Table 14 reports the $\mathcal{F}$ scores obtained with these 2 matching methods, together with their relative deviations. Table 15 further compares the average per-instance processing time of the two matching strategies.

*Table 12.* Scalability results using Qwen-2.5-14B-Instruct as the backbone. We report average logicality and final-answer accuracy on PhysLogic, GPQA, and SciBench.

| Setting | Avg. Logicality | GPQA | SciBench | PhysLogic | Avg. Accuracy |
|---|---|---|---|---|---|
| MegaScience | 43.34 | 33.72 | 41.45 | 40.05 | 38.41 |
| Direct-Distill | 44.60 | 29.07 | 47.15 | 43.06 | 39.76 |
| Logic-Distill | 47.30 | 38.37 | **50.26** | **47.22** | **45.28** |
| RST | **49.85** | **43.02** | 47.15 | 45.60 | 45.26 |

*Table 13.* Out-of-domain results on math benchmarks using Qwen-2.5-7B-Instruct as the backbone. We compare our methods with MegaScience and Direct-Distill baselines.

| Setting | MATH-500 | GSM8K | AIME2025 | AMC | Avg. |
|---|---|---|---|---|---|
| MegaScience | 72.40 | 86.73 | 6.79 | 40.31 | 51.56 |
| Direct-Distill | 71.20 | 84.38 | 6.25 | 41.19 | 50.76 |
| Logic-Distill | **76.00** | **90.45** | **8.54** | **43.15** | **54.54** |
| RST | 74.60 | 88.70 | 8.13 | 40.81 | 53.06 |

*Table 14.* $\mathcal{F}$ scores obtained with greedy matching and dynamic-programming matching, and their relative deviations.

| LLM | Greedy matching | Dynamic-programming matching | Relative deviation |
|---|---|---|---|
| GPT-5 | 54.06 | 55.37 | 2.42% |
| Qwen2.5-7B-Instruct | 57.65 | 59.09 | 2.50% |
| DeepSeek-R1-Distill-Qwen-7B | 47.91 | 48.94 | 2.15% |

*Table 15.* Average per-instance processing time (in seconds) for greedy matching and dynamic-programming matching.

| Matching method | Per-instance time (s) |
|---|---|
| Greedy | 0.1366 |
| Dynamic programming | 1.2403 |

*Table 16.* Logicality scores of the constructed training datasets.

| Dataset | $\mathcal{F}$ | $\mathcal{O}$ | $\mathcal{P}$ |
|---|---|---|---|
| Direct-Distill | 57.70 | 73.28 | 5.29 |
| RST | **63.49** | **77.64** | **7.03** |

Empirically, the dynamic-programming-based scores are highly consistent with those from greedy matching, with relative deviations below $3\%$ and stable model rankings and performance trends under both strategies. At the same time, dynamic programming is nearly an order of magnitude slower than greedy matching. These results indicate that our findings are not sensitive to the specific matching strategy, and that the proposed greedy matching provides a robust and efficient choice for computing logical fidelity.

### D.4. Logicality of the constructed training datasets

To further analyze the properties of our constructed datasets, we compute the three logicality metrics $\mathcal{F}$, $\mathcal{O}$, and $\mathcal{P}$ on the training data of the Direct-Distill and RST settings. Publicly available training corpora are not included in this comparison because they only contain question-answer pairs and do not provide annotated logical nexuses. The results are summarized in Table 16.

We observe that the RST training data consistently achieves higher scores on all three metrics, indicating that it contains reasoning trajectories that are more closely aligned with the expert logical structure. This analysis provides additional evidence that our logic-guided data construction procedure yields training signals with stronger inherent logicality.

*Table 17.* Ablation study (in-domain) under different sampling percentiles in LOGIC-DISTILL (Qwen-2.5-7B-Instruct, all settings trained on 20k examples).

| Data sampled rate of Logic-Distill | $\mathcal{F}$ | $\mathcal{O}$ | $\mathcal{P}$ | Acc |
|---|---|---|---|---|
| 25% | **50.01** | **69.34** | **4.78** | **41.59** |
| 50% | 48.07 | 67.87 | 4.22 | 37.58 |
| 75% | 47.67 | 67.48 | 4.03 | 37.81 |
| 100% (Direct-Distill) | 47.93 | 67.04 | 4.39 | 36.96 |

*Table 18.* Ablation study (out-of-domain) under different sampling percentiles in LOGIC-DISTILL (Qwen-2.5-7B-Instruct, all settings trained on 20k examples).

| Data sampled rate of Logic-Distill | GPQA-D | SciBench | PhysReason |
|---|---|---|---|
| 25% | **32.56** | **35.92** | **40.30** |
| 50% | 29.07 | 34.89 | 36.30 |
| 75% | 26.74 | 24.35 | 27.79 |
| 100% (Direct-Distill) | 24.42 | 19.17 | 17.92 |

### D.5. Ablation on sampling percentiles in "Logic-Distill"

To further examine the effectiveness of our logicality scores, we conduct an ablation study over sampling percentiles in the LOGIC-DISTILL setting. Using Qwen2.5-7B as the backbone, we form three additional LOGIC-DISTILL variants by selecting the top 25%, 50%, and 75% of training examples ranked by their LOGIC-DISTILL scores. For fair comparison, all variants (including the 100% DIRECT-DISTILL baseline) are downsampled to 20k training examples, and evaluated on both in-domain and out-of-domain benchmarks.

Table 17 reports in-domain results on PHYSLOGIC, and Table 18 reports out-of-domain results on GPQA-D, SciBench, and PhysReason. As the sampling threshold is relaxed from top 25% to 100%, performance consistently degrades in both logicality metrics ($\mathcal{F}$, $\mathcal{O}$, $\mathcal{P}$) and task accuracy, indicating that higher LOGIC-DISTILL scores correspond to more valuable training signals. All comparisons above use the same 20k-training checkpoint to control for data scale; nevertheless, data scale also matters for absolute performance (e.g., 40k top-50% generally outperforms 20k top-25%), **motivating our choice of the 50% threshold in the main experiments as a trade-off between logicality and data scale.**

## E. More Details on Dataset Construction

### E.1. Visualization of Data Statistics

Figure 9a illustrates the distribution of the filtered papers across physics subfields, adopting the classification system of the nine major categories for physics on arXiv[13]. Figure 9b presents the distribution of the initial four words within the question sentences. A large number of question lengths and formats highlight the diversity of our constructed dataset.

### E.2. Details of Quality Control

This section provides a detailed introduction to the quality control process, which includes rule-based data filtering, LLM-based quality filtering, and human-based data quality inspection. Specifically:

**Rule-based data filtering** includes:

- **Paper-topic filtering:** We retain only papers with rigorous logical deduction, excluding reviews, tool-development papers, and empirical studies.

- **Forbidden-keyword filtering:** Since the synthesis prompt forbids paper-specific details (e.g., experiments or data), we remove samples whose questions, answers, or logical nexuses contain terms such as "paper," "experimental results," or "author."

---

[13]**astro-ph**: *astrophysics*, **cond-mat**: *condensed matter*, **gr-qc**: *general relativity & quantum cosmology*, **hep**: *high energy physics*, **math-ph**: *mathematical physics*, **nlin**: *nonlinear sciences*, **nucl**: *nuclear physics*, physics: *classical physics*, and **quant-ph**: *quantum physics* (A paper can belong to multiple subfields at the same time).

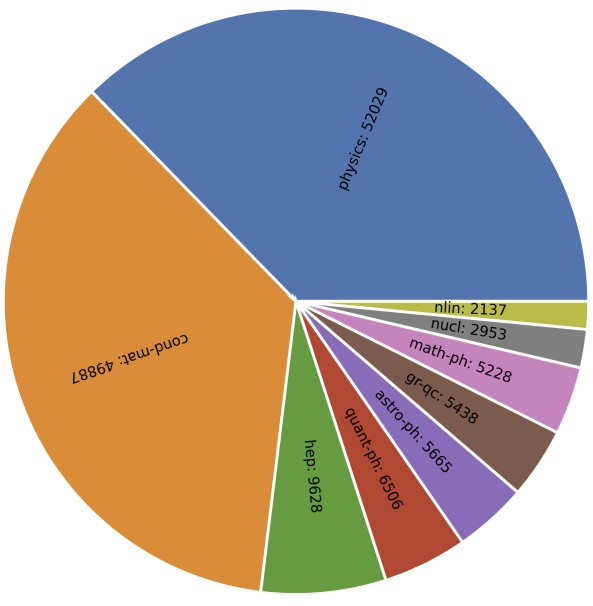

*(a)* Subfield distribution of the filtered papers

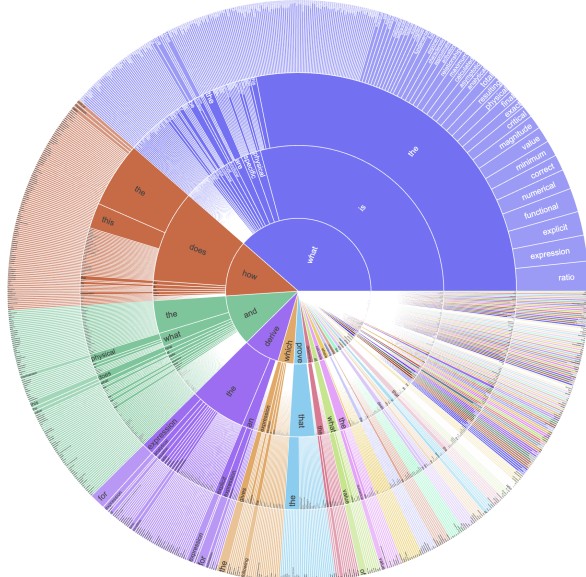

*(b)* Distribution of question quadruplets in the full constructed QA set

*Figure 9.* Visualization of the distribution of the constructed dataset

*Table 19.* The amount of data filtered out at each step of the quality control process.

| Filtering Step | Quantity |
|---|---|
| Initially Collected Papers | 380678 |
| *Rule-based Filtering* | |
| Paper Topic Filtering | 262639 |
| Forbidden Keywords | 1764 |
| Incorrect Formats | 354 |
| Deduplication | 243 |
| *LLM-based Filtering* | |
| Forbidden Keywords | 3258 |
| Data Quality | 1439 |
| **Final Remaining Data** | **110981** |

- **Format filtering:** We discard samples with invalid answer or nexus formats, including malformed multiple-choice items, missing required final-answer formats, or incorrectly formatted logical nexuses.

- **Deduplication:** We apply MinHash LSH and remove near-duplicate questions with Jaccard similarity $> 0.8$.

**LLM-based data filtering** includes:

- **Filtering of forbidden keywords:** With the same objective as the first point above, this step filters out data containing specific content from the paper.

- **Filtering for data quality:** This step filters out data with incomplete information, incorrect question types, or overly simplistic reasoning steps.

The prompt for the LLM-based evaluation is provided in Appendix I.5.

**Human-based data quality inspection:** We randomly sampled 200 data points from the generated dataset, and two Ph.D. students scored each data point against the following four dimensions:

*Table 20.* Human evaluation results on 200 sampled data points. All scores are percentages (%).

| Rater | RP | QQ | AQ | NQ | Average |
|---|---|---|---|---|---|
| Rater 1 | 100.0 | 99.0 | 94.5 | 93.0 | 96.63 |
| Rater 2 | 98.5 | 96.0 | 88.0 | 88.0 | 92.6 |

\* **RP**: Relevance to Paper; **QQ**: Question Quality; **AQ**: Answer Quality; **NQ**: Nexus Quality.

*Table 21.* Consistent scores between the two raters.

| Metric | RP | QQ | AQ | NQ | Average |
|---|---|---|---|---|---|
| Percentage Agreement (%) | 98.5 | 96.0 | 88.0 | 88.0 | 92.6 |
| Brennan-Prediger | 0.97 | 0.92 | 0.76 | 0.76 | 0.85 |

\* **RP**: Relevance to Paper; **QQ**: Question Quality; **AQ**: Answer Quality; **NQ**: Nexus Quality.

**Brennan-Prediger**: used to measure inter-annotator agreement. A value approaching 1.0 indicates near-perfect agreement between raters after correcting for chance.

*Table 22.* Detailed information of the datasets used for training in the experiment.

| Dataset | Data Source | Generation Method | Disciplines | Discipline Labelled | Total Volume | Physics Volume | Sample Ratio |
|---|---|---|---|---|---|---|---|
| Natural Reasoning (Yuan et al., 2025) | Pre-training corpora | LLM-based Synthesis | Physics Computer Science Math Economics Social Sciences | No | 1145824 | - | 0.07 |
| MegaScience (Fan et al., 2025) | University textbooks & public datasets | Corpus Extraction | Medicine Biology Chemistry Computer Science Physics Math Economics | Yes | 1253230 | 41410 | 1.93 |
| Sci-Instruct (Zhang et al., 2024a) | Unlabeled scientific questions | LLM-based Synthesis | Physics Chemistry Math | Yes | 254051 | 123869 | 0.65 |
| SCP-116k (Lu et al., 2025) | Academic documents | Corpus Extraction | Physics Chemistry Biology | Yes | 274166 | 162192 | 0.49 |
| Ours | Academic literatures | LLM-based Synthesis | Physics | Yes | 110981 | 110981 | 0.72 |

*Table 23.* Hyperparameter settings for model inference.

| max_tokens | temperature | top_p | n |
|---|---|---|---|
| 65536 | 0.6 | 0.95 | 8 |

- **Relevance to Paper (RP):** Is the question related to the core research or derivation process of the paper?

- **Question Quality (QQ):** Is the question complete, free of missing information and formatting errors, and does not give the answer away in the question?

- **Answer Quality (AQ):** Is the answer correct?

- **Nexus Quality (NQ):** Does the logical nexus correctly describe the derivation process for this question based on the derivations in the paper?

Table 19 shows the amount of data filtered out by each step of the rule-based and LLM-based filtering. Table 20 presents the average data quality scores for the 200 sampled items across the four dimensions as assessed by the two human raters. Table 21 reports the percentage agreement and Brennan-Prediger score (Brennan & Prediger, 1981) between the two raters.

*Table 24.* Detailed information about the evaluated Closed-Source LLMs

| Model | Version | LLM type |
|---|---|---|
| gpt-5 | - | reasoning |
| gpt-5-mini | - | reasoning |
| gpt-5-nano | - | reasoning |
| o4-mini | - | reasoning |
| doubao-seed-1.6-thinking | 250615 | reasoning |
| claude-3.7-sonnet | 20250219 | reasoning |
| gemini-2.5-flash | preview-04-17 | reasoning |
| grok-4-fast-reasoning | - | reasoning |
| yi-large | - | chat |

*Table 25.* Detailed information about the evaluated Open-Source LLMs

| Model | Version | LLM type | Parameters (B) |
|---|---|---|---|
| DeepSeek-V3 | - | chat | 671 (37B act.) |
| DeepSeek-R1 | - | reasoning | 671 (37B act.) |
| DeepSeek-R1-Distill-Qwen-14B | - | reasoning | 14 |
| DeepSeek-R1-Distill-Qwen-7B | - | reasoning | 7 |
| GLM-4.5 | - | reasoning | 355 (32B act.) |
| Kimi-K2 | 0905 | chat | 1000 (32B act.) |
| Llama-3.1-8B-Instruct | - | chat | 8 |
| Qwen2.5-32B-Instruct | - | chat | 32 |
| Qwen2.5-14B-Instruct | - | chat | 14 |
| Qwen2.5-7B-Instruct | - | chat | 7 |

# F. Implementation Details

## F.1. Details on Training Dataset

Table 22 reports the details of the four public datasets: NaturalReasoning[14], MegaScience[15], Sci-Instruct[16] and SCP-116k[17] and the dataset we constructed used in the training process.

## F.2. Implementation Details on Benchmarking

This section details the evaluation setup on three public benchmarks and PHYSLOGIC benchmark to ensure the reproducibility of our results.

**GPQA:** The evaluation is conducted using **a public third-party framework-ScienceEval**[18]. We test 86 multiple-choice physics questions from the diamond subset. Answer correctness is determined using the framework's rule-based method.

**SciBench:** We also employ the **ScienceEval** framework to evaluate 193 computational physics problems. Answer correctness is verified through a combination of rule-based methods and a mathematical validation library.

**PhysReason:** We utilize **a public third-party framework-Evalscope**[19] for evaluation. We selected plain-text physics problems and decomposed multi-part questions into individual items to facilitate assessment by LLMs. The evaluation uses the framework's custom question-answering pipeline, and answer correctness is determined via the LLM-as-a-judge approach, with `deepseek-v3-0324`[20] serving as the judge LLM.

**PhysLogic:** The complete benchmark, comprising 864 problems, along with the full code for inference, answer assessment, and logicality evaluation, **is provided in the supplementary materials**. We observed that the judge LLM exhibited

---

[14]https://huggingface.co/datasets/facebook/natural_reasoning

[15]https://huggingface.co/datasets/MegaScience/MegaScience

[16]https://huggingface.co/datasets/zd21/SciInstruct

[17]https://huggingface.co/datasets/EricLu/SCP-116K

[18]https://github.com/ScienceOne-AI/ScienceEval

[19]https://github.com/modelscope/evalscope

[20]https://api-docs.deepseek.com/news/news250325

*Table 26.* Sensitivity analysis for the weights of the three logicality dimensions ($\delta_{\mathcal{F}}$, $\delta_{\mathcal{O}}$, $\delta_{\mathcal{P}}$).

| Weight Settings | | | Llama-3.1-8B | | Qwen-2.5-7B-Instruct | | DeepSeek-R1-Distill-Qwen-7B | |
|---|---|---|---|---|---|---|---|---|
| $\delta_{\mathcal{F}}$ | $\delta_{\mathcal{O}}$ | $\delta_{\mathcal{P}}$ | Logicality | Answer | Logicality | Answer | Logicality | Answer |
| 0 | 0.5 | 0.5 | 43.90 | 33.85 | 40.03 | 40.59 | 41.58 | 45.08 |
| 0.5 | 0 | 0.5 | 44.05 | 31.31 | 38.35 | 38.25 | 41.38 | 38.68 |
| 0.5 | 0.5 | 0 | 44.06 | 33.72 | 40.77 | 38.72 | 41.69 | 45.18 |

\* The worst-performing result for each metric is highlighted in red.

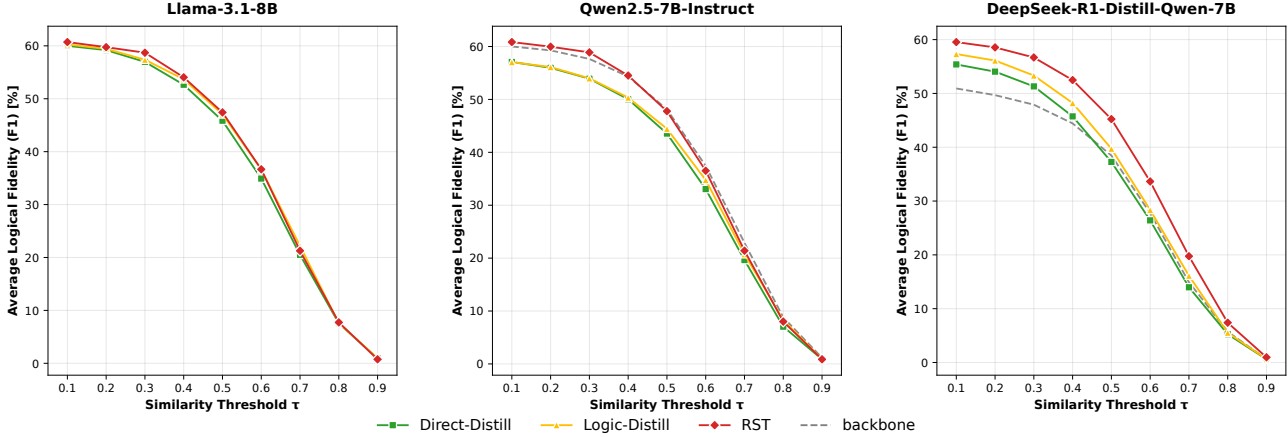

*Figure 10.* Logical fidelity of various models vs. similarity threshold $\tau$

significant variability when evaluating proofs and expression derivation problems. Therefore, to ensure objective and robust answer assessment, we limited our final answer evaluation to the 216 multiple-choice and 216 numerical computation questions. Multiple-choice questions are judged using a rule-based method, while computational questions are assessed using a hybrid of mathematical validation and an LLM judge.

### F.3. Details on LLM Deployment

Model deployment during the evaluation process is facilitated by the `lmdeploy`[21] framework. The specific hyperparameters used for inference are detailed in Table 23. Because some closed-source LLMs do not support some of the parameters set in the main experiment (see Appendix F.2), the experiments on closed-source models only fixed temperature=**0.6**, and the rest of the parameters were not set. Table 24 and 25 summarize the detailed information of the LLMs used in experiment.

## G. Parameter Sensitivity Analysis

### G.1. Analysis on Logicality Dimension Weights

In the Distillation with Logic Supervision process, the score of a sample ($\mathcal{S}$) is calculated as a weighted sum of logical fidelity ($\mathcal{F}$), causal connection ($\mathcal{O}$), and inferential progress ($\mathcal{P}$):

$$\mathcal{S} = \delta_{\mathcal{F}} \cdot \left( 2 \cdot \frac{\text{Norm}(\pi) \cdot \text{Norm}(\rho)}{\text{Norm}(\pi) + \text{Norm}(\rho)} \right) + \delta_{\mathcal{O}} \cdot \text{Norm}(\mathcal{O}) + \delta_{\mathcal{P}} \cdot \text{Norm}(\mathcal{P})$$

We performed a sensitivity analysis to set the final weights for our three logicality dimensions ($\delta_{\mathcal{F}}$, $\delta_{\mathcal{O}}$, and $\delta_{\mathcal{P}}$). In this analysis, we individually removed the influence of each dimension by setting its respective weight to 0 while keeping the other two equal, and then sampled a 40k dataset for training[22]. The results in Table 26 show that removing Causal Connection ($\delta_{\mathcal{O}}$) leads to the most significant performance degradation. We attribute this to the fact that errors in the causal sequence of reasoning are the most critical logical flaws. Therefore, we assigned $\delta_{\mathcal{O}}$ the highest final weight, with the final configuration set to ($\delta_{\mathcal{F}} = \mathbf{0.25}$, $\delta_{\mathcal{O}} = \mathbf{0.50}$, $\delta_{\mathcal{P}} = \mathbf{0.25}$).

---

[21]https://github.com/InternLM/lmdeploy

[22]This experimental setup is the same as that in the ablation study (Section 4.5).

### G.2. Analysis on Similarity Threshold in Logical Fidelity

In the calculation of Logical Fidelity, we employ a similarity threshold, $\tau$, within the greedy matching algorithm. In our main experiments, this threshold was set to **0.3**. However, the choice of $\tau$ is a critical hyperparameter that could influence the evaluation results. To strengthen the credibility and reliability of our evaluation, we examined the effect of varying the similarity threshold. Specifically, we set the threshold $\tau$ to values of $0.1, 0.2, \ldots, 0.9$. We then compared the Logical Fidelity of models trained on our sampled dataset against those trained on a baseline dataset, with this evaluation being conducted across all three backbones.

As shown in Figure 10, the logical fidelity of all tested LLMs decreases with a higher similarity threshold. Our proposed "RST" and "logic-distill" data sampling methods maintain superior performance over the baselines across the entire range of threshold values.

## H. Scoring Rubric for Human Experts and LLM Judge

The following shows the scoring criteria used in Section 4.2 for human experts and LLM judges to assess the logicality of the reasoning process.

---

**Scoring rubric for human experts and LLM judger.**

**You are given** (i) a student's step—by—step solution to a physics problem, and (ii) the reference (gold) final answer together with (iii) a set of key *logicality nexuses* that constitute the essential reasoning structure for solving the problem.
Your task is to evaluate the *logicality* of the student's reasoning trajectory and assign a single overall score on a 1—10 scale.

When scoring, please jointly consider the following three criteria:

— **Logical Fidelity (F):** Are the intermediate claims, formula usages, and deductions consistent with the problem statement and correct physics? Penalize factual/theoretical mistakes, misapplied formulas, or invalid inferences.
— **Logical Overallness (O):** Do the steps form a coherent chain with correct dependency/causal relations that align with the provided key logicality nexuses? Penalize missing links, non sequiturs, contradictions, or a collection of disconnected correct statements.
— **Logical Precision / Progress (P):** Does the reasoning make clear, non—redundant progress toward the final answer with sufficient specificity (e.g., explicit variable definitions, necessary computations, and justified transitions), rather than vague, circular, or hand—wavy statements?

**Score anchors:**

— *1~2 (Severely flawed):* Multiple critical physics/logic errors; steps are largely disconnected or self—contradictory; reasoning does not meaningfully approach the gold answer or the key nexuses.
— *3~4 (Weak):* Some correct fragments but major gaps or unjustified leaps; important nexuses are missing or misused; progress is limited.
— *5~6 (Moderate):* Mostly reasonable but with noticeable inaccuracies, incomplete justifications, or partial coherence; may reach the answer with ambiguity or incomplete derivations.
— *7~8 (Strong):* Largely correct and well—connected chain consistent with the key nexuses; only minor issues/omissions; clear progress with adequate computations/justifications.
— *9~10 (Excellent):* Fully faithful physics, tightly connected reasoning aligned with the key nexuses, and precise step—by—step progress; assumptions are stated and the final answer follows unambiguously.

---

# I. Prompt Design

## I.1. Prompts of Self QA

Below is the prompt to generate the question:

---

**System prompt for question generation (Non-MCP Question)**

For the paper I gave you above, you need to generate a physics exam question from it.

## Guidelines for Generating the Exam Question
Please generate a physics problem based on the core derivation process in the Paper provided above. The problem must be complete and scientifically logical, requiring the examinee to use derivation methods in physics to answer. Specifically, the requirements are as follows:
  – The problem should originate from the core theoretical derivation or proof ideas of the paper.
  – The problem must be specific, clearly defined, with a definite answer. It cannot be ambiguous, subjective, or open–ended.
  – The problem should be self–contained, because the generated question is for an exam, and the examinee cannot read the paper. Therefore, background knowledge, symbol definitions, and other important information mentioned in the paper must be clearly defined in the problem statement. At the same time, the problem cannot rely on the design of methods and experimental parts of the paper, and the problem statement must not contain words such as `author`, `this paper`, `experiment`, etc.
  – Do not provide too many extra thought restrictions in the problem. Do not prompt or restrict the angle or framework of the student's answer. Do not give too many hints in the exam question. Only provide the minimal necessary information to solve the problem. It is strictly forbidden to directly give the core derivation formulas in the question. (Ensure the problem ends with a question mark. After the question, it is strictly forbidden to add extra instructions such as `Please elaborate...`, `Please answer by combining...`, `Please follow...` etc.)
  – The problem must be independent. Do not ask 2 or more questions at the same time. Do not include multiple sub–questions or subtasks in one problem. Do not ask questions in a multi–step manner. (That is, it is strictly forbidden to appear as `Please answer: (1) [Question One] (2) [Question Two] (3) [Question Three]` or `Please answer [Question One]? And further answer [Question Two]?` with multiple questions.)

## Difficulty Requirements
For the difficulty of the generated problem, please follow the requirements below:
  – The problem difficulty should reach the {difficulty} level.
  – The examinee needs to systematically master the corresponding level of the physics knowledge system in order to answer the question correctly.
  – The problem should require multiple steps of derivation, reasoning, or calculation to reach the answer, and the number of steps required must be greater than 5.

## Special Notes
In summary, pay special attention to the following 7 points when generating the problem:
1. **Generate from the core derivation process of the paper**: The problem must come from the paper's core theoretical derivation or proof ideas, and the solution should be consistent with the paper's core derivation ideas.
2. **Ensure the independence of the generated problem**: Only one problem can be asked. It is absolutely forbidden to ask 2 or more questions simultaneously, to include multiple sub–questions or subtasks in one problem, or to give multi–step questioning.
3. **Ensure the nature of an exam question**: This is an exam question. The question must include all the necessary background knowledge and definitions to answer it. The problem must not include any details or references related to the paper's method design and experimental parts.
4. **Ensure the exam nature of assessment**: The problem statement must not give prompts or restrictions on the answering method, answering angle, or thinking framework. Do not give too many extra hints or step prompts. It is also prohibited to directly give key formulas.
5. **{difficulty} level**: The problem difficulty should reach the {difficulty} level. The examinee should need multiple steps of derivation, reasoning, or calculation to obtain the answer, and the number of steps must be greater than 5.
6. **Use LaTeX format for mathematical formulas**: Be sure to write all mathematical symbols, physical formulas, and chemical molecular formulas using standard LaTeX format. Use single dollar signs ($...$) for inline formulas and double dollar signs ($$\n...\n$$) for block formulas. Note: avoid using Unicode characters directly. For complex formulas, ensure clear structure, accurate symbols, and compliance with LaTeX typesetting standards.
7. **Only output the problem, not the answer**: You only need to provide the exam problem. You do not need to provide the reference answer.

## Output Format
Please output the question directly as **plain English text**, without using code blocks, JSON, or any other formatting methods, and do not include any prefix or suffix.

---

**System prompt for question generation (MCP Question)**

For the paper I gave you above, you need to generate a physics multiple–choice exam question with exactly four options (A–D).

## Guidelines for Generating the Exam Question
Please generate a physics multiple–choice problem based on the core derivation process in the Paper provided above. The problem must be complete and scientifically logical, requiring the examinee to use derivation methods in physics to answer. Specifically, the requirements are as follows:
  – The problem should originate from the core theoretical derivation or proof ideas of the paper.
  – The problem must be specific, clearly defined, with a definite answer. It cannot be ambiguous, subjective, or open–ended.

– The problem should be self—contained, because the generated question is for an exam, and the examinee cannot read the paper. Therefore, background knowledge, symbol definitions, and other important information mentioned in the paper must be clearly defined in the problem statement. At the same time, the problem cannot rely on the design of methods and experimental parts of the paper, and the problem statement must not contain words such as `author`, `this paper`, `experiment`, etc.

– Do not provide too many extra thought restrictions in the problem. Do not prompt or restrict the angle or framework of the student's answer. Do not give too many hints in the exam question. Only provide the minimal necessary information to solve the problem. It is strictly forbidden to directly give the core derivation formulas in the question. (Ensure the problem ends with a question mark. After the question, it is strictly forbidden to add extra instructions such as `Please elaborate...`, `Please answer by combining...`, `Please follow...` etc.)

– The problem must be independent. Do not ask 2 or more questions at the same time. Do not include multiple sub—questions or subtasks in one problem. Do not ask questions in a multi—step manner. (That is, it is strictly forbidden to appear as `Please answer: (1) [Question One] (2) [Question Two] (3) [Question Three]` or `Please answer [Question One]? And further answer [Question Two]?` with multiple questions.)

– The output must consist of one stem ending with a question mark, followed by exactly four options, each on its own line starting with `A.`, `B.`, `C.`, and `D.` respectively.

– Ensure exactly one unambiguously correct option and three plausible distractors. Distractors should reflect typical physics mistakes (e.g., sign error, missing factor, wrong boundary condition/limit, units mismatch, or misuse of an approximation) rather than being obviously wrong. Keep option length and style comparable; avoid hedging words that make one option stand out.

– The correct option must correspond to the result obtained from the full multi—step derivation (>5 steps). Each distractor should correspond to a specific, common derivation slip (e.g., dropping a term, wrong limit, factor—of—2 error), not to arbitrary values.

## Difficulty Requirements
For the difficulty of the generated problem, please follow the requirements below:
  – The problem difficulty should reach the {difficulty} level.
  – The examinee needs to systematically master the corresponding level of the physics knowledge system in order to answer the question correctly.
  – The problem should require multiple steps of derivation, reasoning, or calculation to reach the answer, and the number of steps required must be greater than 5.

## Special Notes
In summary, pay special attention to the following 7 points when generating the problem:
1. **Generate from the core derivation process of the paper**: The problem must come from the paper's core theoretical derivation or proof ideas, and the solution should be consistent with the paper's core derivation ideas.
2. **Ensure the independence of the generated problem**: Only one problem can be asked. It is absolutely forbidden to ask 2 or more questions simultaneously, to include multiple sub—questions or subtasks in one problem, or to give multi—step questioning.
3. **Ensure the nature of an exam question**: This is an exam question. The question must include all the necessary background knowledge and definitions to answer it. The problem must not include any details or references related to the paper's method design and experimental parts.
4. **Ensure the exam nature of assessment**: The problem statement must not give prompts or restrictions on the answering method, answering angle, or thinking framework. Do not give too many extra hints or step prompts. It is also prohibited to directly give key formulas.
5. **{difficulty} level**: The problem difficulty should reach the {difficulty} level. The examinee should need multiple steps of derivation, reasoning, or calculation to obtain the answer, and the number of steps must be greater than 5.
6. **Use LaTeX format for mathematical formulas**: Be sure to write all mathematical symbols, physical formulas, and chemical molecular formulas using standard LaTeX format. Use single dollar signs ($...$) for inline formulas and double dollar signs ($$\n...\n$$) for block formulas. Note: avoid using Unicode characters directly. For complex formulas, ensure clear structure, accurate symbols, and compliance with LaTeX typesetting standards.
7. **Only output the problem, not the answer**: You only need to provide the exam problem. You do not need to provide the reference answer. Do not indicate which option is correct (e.g., no ticks, boldface, parentheses, or phrases like 'Correct answer: B').

## Output Format
Please output the question with four choices directly as **plain English text**, with four options following the stem, each on its own line starting with `A.`, `B.`, `C.`, and `D.` respectively. Do not use code blocks, JSON, or any other formatting methods, and do not include any prefix or suffix.

Below is the prompt to generate the answer:

## System prompt for answer generation

Now, for the above question, please provide a reference answer.

## Guidelines for Generating the Answer
Please answer the exam question you just posed by following strict scientific logic, with the requirements:
  – The answer must adhere to scientific logic, organize and recall the background knowledge used in the question, and complete the solution through rigorous multi—step reasoning.
  – The answer should refer to the core derivation ideas in the paper.
  – The answer should be descriptive, using the details and symbols defined in the question.
  – Because this is an exam question, the solver does not have access to this paper, so the answer must not rely on the paper's data, experimental results, or other specific details, and cannot cite external resources such as figures, tables, or videos. Ensure that the solver can fully understand the solution by only reading the question and the answer.
  – Similarly, avoid using words such as `author`, `this paper`, or `experiment` in the answer.

## Mathematical Formula Rules
All mathematical symbols, physical formulas, chemical formulas, etc. must be written in standard LaTeX formula format, using single dollar signs ($...$) for inline formulas, and double dollar signs ($$
...
$$) for display formulas. Note that Unicode characters should be avoided. For complex formulas, ensure that the structure is clear, the symbols are accurate, and the typesetting follows LaTeX mathematical conventions.

## Output Format
Please output the answer directly as **plain English text**, avoiding the use of code blocks, JSON, or other formatting methods. No prefix or suffix is needed. **If the question is a calculation−type problem with a final solution or expression, express the final answer as a decimal number with three digits after the decimal point. Conclude the answer by stating "The answer is therefore \boxed{[ANSWER]}.**

## I.2. Prompts of Inference

Below is the prompt for a strong LLM to answer the question directly:

### Prompt for answer question directly (Non-MCP Question)

— Please answer this question: [question]
— If the problem requires a numerical calculation or yields a final numerical expression, express the final answer as a decimal number with three digits after the decimal point.
— Provide a step−by−step reasoning process, then at the end state the final answer in `\boxed{}`.

### Prompt for answer question directly (MCP Question)

— Please answer this question: [question]
— Provide a step−by−step reasoning process, then at the end state the final answer in `\boxed{}` (enter only a single option letter, e.g., `\boxed{A}`, do not include the full option text).

Below is the prompt for strong LLM to transfer the logical nexuses into a continuous reasoning process:

### Prompt for style transfer

## Task Background
— You are a physics expert. Above I have provided you with three pieces of text related to a physics problem. The first text 1. Exam Question presents a physics problem. The second text 2. Key Solution Process of the Reference Answer contains some key scoring points for solving the problem, with highly scientific and correct content. The third text 3. Human Problem−Solving Thought Process records, in the first person, the thought process of a human volunteer solving another problem. This may include (but is not limited to) problem decomposition, recalling known background knowledge, classification and exploration, logical reasoning, mathematical derivation, self−questioning, self−reflection and verification, summary and induction, etc., which represent the paradigms of human scientific reasoning. The language style is more natural, and the logical transitions between solution steps are also more coherent.

## Task Objective
— **Your task is to reorganize and polish 2. Key Solution Process of the Reference Answer, transforming it into a natural reasoning process in the style of 3. Human Problem−Solving Thought Process. The polished text should maintain the scientific validity and correctness of the original solution, while reflecting the natural, coherent, and logical thinking and language style of a human solving the problem.**

## Guidelines for Style Conversion of Scientific Logical Reasoning
— **Ensure the reasoning follows human problem−solving style**: Your polished reasoning process must **include (but is not limited to) problem decomposition, recalling known background knowledge, classification and exploration, logical reasoning, mathematical derivation, self−questioning, self−reflection and verification, summary and induction, etc.** It should be expressed in the form of a first−person narrative, showing the solver's specific thoughts at each step while solving the problem along the lines of 2. Key Solution Process of the Reference Answer. Specifically, you may imitate 3. Human Problem−Solving Thought Process to clearly show internal thoughts like: "Hmm, this problem looks complicated, let me analyze it step by step...", "Let me recall...", "Now the issue is..., I can solve it using... method", "Let me think, is it possible that...?", etc.
— **Ensure completeness of reasoning steps and logical coherence**: 2. Key Solution Process of the Reference Answer only summarizes the key steps necessary for solving the problem, but the logical transitions are abrupt, and some steps may be skipped. Therefore, you need to supplement necessary intermediate steps to make the entire reasoning chain more complete — not just "knowing what to do," but also "understanding why to do so," and making the motivation behind each step clear. In addition, please make the logical transitions between steps explicit, so that the derivation process flows more naturally and smoothly, avoiding a mechanical style with numbered structures such as "Step 1, Step 2..." or "1., 2., ...".
— **Preserve formulas**: Formulas are very important. Do not remove any formulas from 2. Key Solution Process of the Reference Answer. Ensure the formulas are clear in structure, symbols are accurate, and each formula's meaning and motivation for use are fully explained.
— **Representation of symbols in formulas**: For formatting consistency, in the polished reasoning process, please use Unicode characters to represent operators, mathematical symbols, subscripts, superscripts, and Greek letters, instead of LaTeX format. If 2. Key Solution Process of the Reference Answer uses LaTeX format for formulas, please convert them into Unicode

character format.

## Output Format
Please output the polished scientific logical reasoning directly as English text, avoiding code blocks or JSON formatting. The output should be continuous text, without leading phrases like "The answer is as follows" or "The answer is," and without concluding summaries, to ensure a clean format that is ready for direct use.

## I.3. Prompts of Logical Nexuses Extraction

Below is the prompt to extract logical nexuses from the paper:

**System prompt to extract logical nexuses**

I have provided you with an exam question above, along with its reference answer and a related reference paper. Your task is to summarize, in a concise way and following the logical order of derivation, several key points of the reasoning/derivation/calculation process required to solve this question, which will serve as the scoring points for grading the solution process.

# Guidelines for Generating Scoring Points in the Solution Process of a Scientific Exam Question
Please follow these requirements:
  — This scientific exam question is highly related to the core scientific problem addressed in the reference paper, therefore the solution process should be guided by the reasoning ideas from the paper.
  — The reference paper is only used as a reference for the core reasoning approach, because students cannot see the paper during the exam. Therefore, in the scoring points you summarize, avoid mentioning experimental results or overly specific details of the paper, and do not use words such as "this paper", "in the text", "author", "experimental conclusion", "experimental analysis", etc.
  — The scoring points should clearly capture the process of reasoning step by step from the start of the problem to the final answer.
  — The scoring points should be highly concise and clearly expressed, avoiding ambiguity, so that graders can use them to make definite judgments on the students' answers.
  — The scoring points should be connected naturally and logically, avoiding missing or skipping steps. Moreover, the sequence of the scoring points must reflect a correct logical order, not reversed.
  — The number of scoring points should be between 5 and 15, determined according to the key steps of reasoning in the paper's core ideas.
  — Based on the importance of each scoring point, you should assign appropriate scores to them, with the total score being 100.

# Example
For a scoring point, the format can refer to the example below:
  1. Calculate the difference in coefficients of thermal expansion: $\alpha_{\text{torsion}} - \alpha_{\text{sub}} = 3.5 \times 10^{-6} \,^{\circ}\text{C}^{-1}$ (10 points)

# Output Format
Please output these scoring points directly in English text, one point per line, each starting with an ordered list number (e.g., "1.") and ending with the score in parentheses "(x points)". Avoid using code blocks or JSON formatting, and do not include any prefixes or suffixes.

## I.4. Prompts of Benchmarking

Below is the prompt for GPQA benchmark's evaluation:

**GPQA prompt**

{question}
Please wrap the choice of answer at the end of the solving process using `\\boxed{{}}` to highlight it.

Below is the prompt for SciBench benchmark's evaluation:

**SciBench prompt**

{question}
For the given scientific problem in the Chemistry, Physics, or Mathematics category, please provide a concise and step−by−step solution. Ensure that your final answer is placed within \box{}. If the final answer contains \pi or \sqrt{2}, convert it to a decimal approximation and calculate the final numerical value, using approximately 3.14159265359 for \pi and 1.41421356237 for \sqrt{2}. If the answer is in scientific notation, such as 1.8 \times 10^4, the base value (10^4) will be provided in the problem, and the final answer should be expressed as 1.8. The unit will be provided in the problem and should not be included in the answer. The final answer should be expressed solely as a decimal number.

Below is the prompt for PhysReason benchmark's evaluation:

---

**PhysReason's inference prompt**

{question}
Please answer this question: {query}.
Please wrap the final result at the end of the derivation process using `\boxed{}` to highlight it.

---

Below is the prompt for judging by LLM during PhysReason evaluating.

---

**PhysReason's judging prompt**

Your job is to look at a question, a gold target, and a predicted answer, and return a letter "A" or "B" to indicate whether the predicted answer is correct or incorrect.

Question: {question}
Reference Answer: {gold}
Model Answer: {pred}

Evaluate the model's answer based on correctness compared to the reference answer.
Grade the predicted answer of this new question as one of:

A: CORRECT
B: INCORRECT

Just return the letters "A" or "B", with no text around it.

---

Below is the prompt for our proposed PHYSLOGIC benchmark's evaluation:

---

**PhysLogic's inference prompt**

Please wrap the choice of answer at the end of the solving process using `\\boxed{{}}` to highlight it.

---

Below is the prompt for LLM judgment during PHYSLOGIC evaluation.

---

**PhysLogic's judging prompt**

Your job is to look at a question, a gold target, and a predicted answer, and return a letter "A" or "B" to indicate whether the predicted answer is correct or incorrect.

[Question]
{question}

[Reference Answer]
{gold}

[Predicted Answer]
{pred}

**Evaluation Rules:**
— The final numerical answer should be the primary focus.
— Minor differences in numerical values due to rounding or variations in significant figures are acceptable.
— A small margin of error (within 5%) is allowed. For example, if the reference answer is 100, predicted answers between 95 and 105 are considered correct.
— If the final answer is an expression rather than a numerical solution, carefully determine whether the two expressions are equivalent.

Grade the predicted answer of this new question as one of:
A: CORRECT
B: INCORRECT

Directly provide your final judgment by putting the option in `\boxed{}`, for example: `\boxed{A}` or `\boxed{B}`.

---

## I.5. Prompts of Quality Control

Below is the prompt for paper topic filtering:

---

**System prompt for paper topic filtering**

You are a researcher. For the paper provided above, your task is to evaluate its scientific originality in terms of theoretical contribution. The goal is to identify physics papers that contain rigorous theoretical derivations while filtering out those that are mainly engineering implementations or survey—type works. Specifically, follow the scoring guidelines below:

## Guidelines for Evaluating Scientific Originality of the Paper
— If the paper proposes a new conjecture for an existing physics problem and verifies it through theoretical derivation, assign a score of **1**.
— If the paper introduces a new method or model for an existing task and proves its validity through theoretical derivation, assign a score of **1**.
— If the paper designs a new physical model for a specific physical scenario and demonstrates its correctness through theoretical derivation, assign a score of **1**.
— If the paper merely designs experiments to observe certain physical phenomena, but lacks substantial theoretical derivation—i.e., its contribution lies mainly in experimental design with little theoretical innovation—assign a score of **0**.
— If the paper only develops a tool through software engineering or design based on existing theories, with its contribution lying mainly in tool construction but lacking methodological innovation, assign a score of **0**.
— If the paper is a review or commentary that surveys or summarizes an existing research area without proposing new scientific innovations, assign a score of **0**.
— If the key methodological or theoretical derivation sections of the paper are missing, corrupted, or otherwise unreadable such that you cannot reasonably understand the content, assign a score of **0**.

Based on the above **Guidelines for Evaluating Scientific Originality of the Paper**, determine whether the score of this paper should be **"1"** or **"0"**, and directly output your final score inside `\boxed{}` — for example, `\boxed{1}` or `\boxed{0}`.

---

Below is the prompt for filtering out data samples with forbidden keywords:

---

**System prompt for forbidden keywords filtering**

Your task is to act as a quality control evaluator. You will be given a question and its corresponding answer. You must determine if the pair adheres to the principle of being "self—contained" for an exam setting. Analyze both the question and the answer based on the rules below and provide a binary judgment.

[Question to be Evaluated]
{question}

[Answer to be Evaluated]
{answer}

**Evaluation Criteria:**

1. **All—inclusive Content**: The question must be entirely self—contained. All necessary context, background information, and definitions for symbols or terms must be included within the question itself. An examinee should not need any external document to understand and solve the problem.
2. **Independence from Source Material**: The question and the answer must not rely on specific methodologies, experimental setups, or results from an external paper. The entire problem and its solution must stand on their own.
3. **Forbidden Keywords**: Neither the question nor the answer should contain words that explicitly refer to a source document or its authors, such as `this paper`, `the author`, `in our experiment`, `the article`, etc.

**Final Judgment:**

Based on your analysis of the rules above, determine if the question and answer pair is compliant.
— **1**: The pair is compliant and meets all the requirements.
— **0**: The pair violates one or more of the requirements.

Directly provide your final judgment by putting the option in `\boxed{}`. For example: `\boxed{1}` for a compliant pair or `\boxed{0}` for a non—compliant one.

---

Below is the prompt for filtering out data samples with incomplete information, incorrect question types, or overly simplistic reasoning steps.

---

**System prompt for low-quality datas filtering**

Your task is to act as a data quality analyst. You will be provided with a question, its answer, and its designated question type. Your goal is to determine if the data is of sufficient quality and correctness based on the criteria below.

[Question]

---

{question}

[Answer]
{answer}

[Asserted Question Type]
{question_type}

**Evaluation Criteria:**

1. **Completeness**: The question must contain all the necessary information, values, and context needed to arrive at the answer. The answer should also be complete and fully address the question asked. Data should be filtered out if the question is unanswerable due to missing details or if the answer is unfinished or truncated.
2. **Correct Question Type**: The content and format of the question must be consistent with the provided `[Asserted Question Type]`. For instance, if the type is "Multiple Choice," the question must actually present options to choose from. If the type is "Numeric Computation," the question should ask for a numerical result.
3. **Sufficient Complexity**: The problem should require non—trivial reasoning or calculation. The reasoning steps, as demonstrated or implied in the answer, should not be overly simplistic. Filter out basic definitional questions or simple one—step arithmetic problems that do not require logical deduction or domain—specific knowledge.

**Final Judgment:**

Based on your analysis of the rules above, determine if the data is of sufficient quality.
— **1**: The data is of good quality and passes all checks.
— **0**: The data is of poor quality and fails one or more checks.

Directly provide your final judgment by putting the option in `\boxed{}`. For example: `\boxed{1}` for good quality data or `\boxed{0}` for poor quality data.

# J. Data Examples

## J.1. Multiple Choice Problem

Below is an example of a multiple choice problem:

- **Difficulty:** High School

- **Subdomain:** *general relativity & quantum cosmology*

---

### Question (Multiple Choice Problem)

Consider a constant-density star (Schwarzschild star) with compactness $C \equiv M/R$ approaching the black hole limit $C \to 1/2$ from below. Define $y_1 = \sqrt{1 - 2C}$ and the coordinate transformation $x = 1 - y = 1 - \sqrt{1 - (r/\alpha)^2}$ where $\alpha = R^{3/2}/\sqrt{R_S}$ and $R_S = 2M$. The interior pressure becomes negative when $C > 4/9$, creating a singular surface at $x_0 = -\kappa$ where $\kappa = 3y_1 - 1$. The tidal Love number $k_2$ quantifies the quadrupolar tidal deformability of this star as measured at its surface $r = R$ ($x = x_1$). When deriving the behavior of $k_2$ near the black hole limit ($C \to 1/2^+$, $\kappa \to 0^-$), what functional dependence on compactness $\delta C = C - 1/2$ does $k_2$ exhibit?
A. $k_2 \propto \delta C$
B. $k_2 \propto 1/\log(|\delta C|)$
C. $k_2 \propto \exp\left(-b/|\delta C|\right)$
D. $k_2 \propto \exp\left(-b|\delta C|\right)$

---

### Answer (Multiple Choice Problem)

To answer this question, we must derive the behavior of the tidal Love number $k_2$ for a Schwarzschild star as the compactness $C \equiv M/R$ approaches the black hole limit $C \to 1/2^-$. The compactness deviation is $\delta C = C - 1/2 < 0$, and we focus on the magnitude $|\delta C| = 1/2 - C$.

The key parameter is $y_1 = \sqrt{1 - 2C}$, which relates to the coordinate transformation $x = 1 - y$, with $y = \sqrt{1 - (r/\alpha)^2}$ and $\alpha = R^{3/2}/\sqrt{R_S}$, where $R_S = 2M$. The compactness $C > 4/9$ (beyond Buchdahl) leads to negative pressure in the interior, with a singular surface at $x_0 = -\kappa$, where $\kappa = 3y_1 - 1$.

As $C \to 1/2^-$, $y_1 \to 0^+$, and $\kappa \to -1^+$. Thus, $x_0 = -\kappa \to 1^-$ and the surface $x_1 = 1 - y_1 \to 1^-$. The difference $|x_1 - x_0| = 2y_1 = 2\sqrt{1 - 2C} \propto \sqrt{|\delta C|}$, so $|x_1 - x_0|$ scales with $\sqrt{|\delta C|}$.

The tidal Love number $k_2$ is derived from the perturbation equation for the logarithmic derivative $h(r)$ (or $H(r)$), governed by a Riccati equation in the $x$-coordinate, with coefficients depending on the metric and pressure. The pressure singularity at $x_0$ introduces essential singular behavior in the perturbation solution. Integrating the equation from $x = 0$ to $x = x_1$ with boundary conditions $h \approx 2$ at small $r$ and continuity at $r = R$, the solution near $x_0$ involves Frobenius expansion or WKB-like methods due to the singularity.

Analysis shows that as $|x_1 - x_0| \propto \sqrt{|\delta C|} \to 0$, the wavefunction solution exhibits exponential suppression proportional to $\exp(-b/\sqrt{|\delta C|})$ for some constant $b > 0$. To match the boundary condition at $x_1$, the value $h_R$ inherited from this solution gives $k_2$ through the formula $k_2 \propto [2C(h_R - 1) - h_R + 2]/$denominator, simplified via the exponential decay. Thus, the dominant behavior is $k_2 \propto \exp(-b/|\delta C|)$ for a constant $b > 0$.

This is contrasted with ECO logarithmic scaling ($k_2 \propto 1/\log|\delta C|$) and linear scaling ($k_2 \propto |\delta C|$), both excluded as they do not match the derived exponential form. Option D ($\exp(-b|\delta C|)$) is invalid as it approaches 1, not 0, when $|\delta C| \to 0$.

The answer is therefore $\boxed{C}$. $\boxed{C}$

---

### Logical Nexus (Multiple Choice Problem)

1. Define the deviation $\delta C = C - 1/2$ and identify $y_1 = \sqrt{1 - 2C} \approx \sqrt{|2\delta C|}$ near the BH limit. (10 points)

2. Express $\kappa = 3y_1 - 1 \approx -1 + 3\sqrt{2|\delta C|}$ as $C \to 1/2^-$, with $\kappa \to -1^+$. (10 points)

3. Apply coordinate transformation $x = 1 - y$ to find the singular surface $x_0 = -\kappa \approx 1 - 3\sqrt{2|\delta C|}$ and surface coordinate $x_1 = 1 - y_1 \approx 1 - \sqrt{2|\delta C|}$. (15 points)

4. Show $|x_1 - x_0| = 2y_1 \approx 2\sqrt{2|\delta C|} \propto \sqrt{|\delta C|} \to 0$ as $\delta C \to 0^-$. (15 points)

5. Formulate the tidal perturbation as a Riccati equation in the $x$-coordinate, noting coefficient singularities at $x_0$ due to pressure divergence. (15 points)

6. Derive the solution's exponential suppression near $x_0$: $\propto \exp(-b/\sqrt{|\delta C|})$ for $b > 0$, using WKB-like asymptotics or Frobenius analysis. (20 points)

7. Evaluate $h$ at the surface ($x_1$) and substitute into the $k_2$ formula to confirm $k_2 \propto \exp(-b/|\delta C|)$, rejecting options A, B, and D. (15 points)

## J.2. Expression Computation

Below is an example of an expression computation problem:

- **Difficulty:** PhD student

- **Subdomain:** *mathematical physics*

---

**Question (Expression Computation Problem)**

In the context of the random loop model on a $d$-dimensional hypercubic lattice, consider the nearest-neighbor connection probability at the same time, denoted as $\kappa$. This probability satisfies the inequality:

$$1 - \kappa \le \theta \frac{I^{u,d}_{(1,-1)}}{\sqrt{2}} \sqrt{\kappa} + \frac{\theta}{4d\beta}$$

where:

- $\theta \in \mathbb{N}$, $\theta \ge 2$ is the number of loop colors,

- $u \in [0, 1/2]$ is the loop model parameter,

- $\beta > 0$ is the inverse temperature satisfying $\beta \ge \theta/(4d)$,

- $d \in \mathbb{N}$ is the lattice dimension,

- $I^{u,d}_{(1,-1)}$ is a constant defined as:

$$I^{u,d}_{(1,-1)} = \sup_{\alpha \in [0,1]} \int_{[0,2\pi]^d} \frac{d^d k}{(2\pi)^d} \sqrt{u\alpha + (1-u)(1-\alpha)\frac{\epsilon(k+\pi)}{\epsilon(k)}} \left(1 - \frac{1}{d}\sum_{j=1}^{d} \cos(k_j)\right)_+$$

with $\epsilon(k) = 2\sum_{j=1}^{d}(1 - \cos k_j)$ and $k + \pi = (k_1 + \pi, \ldots, k_d + \pi)$.

Derive an expression for the lower bound of $\kappa$ in terms of the given parameters $\theta$, $I^{u,d}_{(1,-1)}$, $\beta$, and $d$.

---

**Answer (Expression Computation Problem)**

The inequality provided is:

$$1 - \kappa \le \theta \frac{I^{u,d}_{(1,-1)}}{\sqrt{2}} \sqrt{\kappa} + \frac{\theta}{4d\beta}$$

where $\kappa = \mathbb{P}_{\beta,\theta,u}(E_{0,e,0})$. We solve for a lower bound on $\kappa$ as follows:
1. Rearrange terms to isolate the square root:

$$1 - \frac{\theta}{4d\beta} - \kappa \le \theta \frac{I^{u,d}_{(1,-1)}}{\sqrt{2}} \sqrt{\kappa}$$

2. Define substitutions: Let $x = \sqrt{\kappa}$ (so $x \ge 0$), and set:

$$A = \theta \frac{I^{u,d}_{(1,-1)}}{\sqrt{2}}, \quad B = 1 - \frac{\theta}{4d\beta}$$

The inequality becomes:

$$B - x^2 \le Ax$$

3. Form a quadratic inequality: Rearrange to:

$$x^2 + Ax - B \ge 0$$

This quadratic inequality holds when $x \ge \frac{-A+\sqrt{A^2+4B}}{2}$ (considering $x \ge 0$ and the quadratic's positive root).
4. Substitute back: Since $\kappa = x^2$, the lower bound is:

$$\kappa \ge \left(\frac{-A + \sqrt{A^2 + 4B}}{2}\right)^2$$

Replacing $A$ and $B$:

$$\kappa \ge \left(\frac{-\theta \frac{I^{u,d}_{(1,-1)}}{\sqrt{2}} + \sqrt{\left(\theta \frac{I^{u,d}_{(1,-1)}}{\sqrt{2}}\right)^2 + 4\left(1 - \frac{\theta}{4d\beta}\right)}}{2}\right)^2$$

5. Simplify the expression: The term inside the square root simplifies as:

$$\left(\theta\frac{I^{u,d}_{(1,-1)}}{\sqrt{2}}\right)^2 + 4 - \frac{\theta}{d\beta} = \frac{\theta^2(I^{u,d}_{(1,-1)})^2}{2} + 4 - \frac{\theta}{d\beta}$$

Thus:

$$\kappa \geq \left(\frac{\sqrt{\frac{\theta^2(I^{u,d}_{(1,-1)})^2}{2} + 4 - \frac{\theta}{d\beta}} - \theta\frac{I^{u,d}_{(1,-1)}}{\sqrt{2}}}{2}\right)^2$$

This is the lower bound for the nearest-neighbor connection probability $\kappa$.

$$\kappa \geq \left(\frac{\sqrt{\frac{\theta^2\left(I^{u,d}_{(1,-1)}\right)^2}{2} + 4 - \frac{\theta}{d\beta}} - \theta\frac{I^{u,d}_{(1,-1)}}{\sqrt{2}}}{2}\right)^2$$

## Logical Nexus (Expression Computation Problem)

1. Rearrange the given inequality to isolate constant and $\kappa$ terms: move $\frac{\theta}{4d\beta}$ to the left and $\kappa$ to the right, yielding $1 - \kappa - \frac{\theta}{4d\beta} \leq \theta\frac{I^{u,d}_{(1,-1)}}{\sqrt{2}}\sqrt{\kappa}$. (10 points)

2. Substitute $x = \sqrt{\kappa}$ and define constants: $A = \theta\frac{I^{u,d}_{(1,-1)}}{\sqrt{2}}$ and $B = 1 - \frac{\theta}{4d\beta}$, transforming the inequality to $B - x^2 \leq Ax$. (20 points)

3. Rearrange the substituted inequality into standard quadratic form: $x^2 + Ax - B \geq 0$. (10 points)

4. Solve the quadratic inequality by identifying the relevant root for $x \geq 0$: $x \geq \frac{-A+\sqrt{A^2+4B}}{2}$. (30 points)

5. Substitute $\kappa = x^2$ back into the solution, yielding $\kappa \geq \left(\frac{-A+\sqrt{A^2+4B}}{2}\right)^2$. (10 points)

6. Replace $A$ and $B$ with their expressions and simplify the square root term to $\sqrt{\frac{\theta^2(I^{u,d}_{(1,-1)})^2}{2} + 4 - \frac{\theta}{d\beta}}$. (10 points)

7. Write the final expression for the lower bound of $\kappa$ using the simplified terms. (10 points)

## J.3. Numeric Computation

Below is an example of a numeric computation problem:

- **Difficulty:** Master's student

- **Subdomain:** *classical physics*, *condensed matter*

---

### Question (Numeric Computation Problem)

In high-harmonic interferometry studies of solids, a phase shift $\Delta\phi$ in the harmonic radiation can arise from excitation-induced bandgap changes. The dipole phase for a harmonic of order $N$ is given by $\phi = N(\omega_0 t_r + \pi/2) - S(t_r)$, where $S(t_r) = \int_{t_i}^{t_r} \Delta\varepsilon(k(\tau))d\tau$ is the semi-classical action in atomic units, $t_i$ and $t_r$ are the excitation and recombination times, and $\Delta\varepsilon$ is the energy difference between bands.

Suppose an excitation uniformly increases $\Delta\varepsilon$ by a constant $\Delta E_g$ (in hartree) across all $k$. Using the semi-classical model, derive the expression for $\Delta\phi = \phi' - \phi$, where $\phi'$ is the phase after the perturbation.

In an experiment, the phase shift of the 7th harmonic ($N = 7$) is measured as $-1.2$ radians, with a characteristic excursion time of 1.5 femtoseconds. Using the conversion factors 1 fs $= 10^{-15}$ s, 1 a.u. time $= 2.4188 \times 10^{-17}$ s, and 1 hartree $= 27.211$ eV, calculate $\Delta E_g$ in eV.

---

### Answer (Numeric Computation Problem)

The phase shift $\Delta\phi$ is related to the bandgap change $\Delta E_g$ and the characteristic excursion time $\Delta t$ by the formula $\Delta\phi = -\Delta E_g \Delta t$, where $\Delta\phi$ is in radians, $\Delta E_g$ is in hartree, and $\Delta t$ is in atomic units of time. Given $\Delta\phi = -1.2$ radians and $\Delta t = 1.5$ femtoseconds, first convert $\Delta t$ to atomic units using the provided conversions.

1 femtosecond (fs) $= 10^{-15}$ seconds, and 1 atomic unit (a.u.) of time $= 2.4188 \times 10^{-17}$ seconds. Thus:

$$\Delta t_{\mathrm{au}} = \frac{1.5 \times 10^{-15}}{2.4188 \times 10^{-17}} = \frac{1.5}{2.4188} \times 10^2 \approx 62.0142267 \text{ a.u. time.}$$

Solve for $\Delta E_g$ in hartree:

$$-1.2 = -\Delta E_g \times 62.0142267 \implies \Delta E_g = \frac{1.2}{62.0142267} \approx 0.01935239 \text{ hartree.}$$

Convert $\Delta E_g$ to electron volts using 1 hartree $= 27.211$ eV:

$$\Delta E_g \text{ (in eV)} = 0.01935239 \times 27.211 \approx 0.52659788429 \text{ eV.}$$

Rounding to three decimal places: 0.52659788429 rounds to 0.527 eV, as the fourth decimal place is 5 (followed by 9), requiring rounding up.

The answer is therefore $\boxed{0.527}$.

---

### Logical Nexus (Numeric Computation Problem)

1. Recognize that a uniform increase in $\Delta E_g$ modifies the semi-classical action to $S'(t_r) = S(t_r) + \Delta E_g(t_r - t_i)$. (10 points)

2. Express the perturbed dipole phase as $\phi' = N(\omega_0 t_r + \pi/2) - [S(t_r) + \Delta E_g(t_r - t_i)]$. (10 points)

3. Formulate the phase shift $\Delta\phi = \phi' - \phi = -[S'(t_r) - S(t_r)] = -\Delta E_g(t_r - t_i)$. (15 points)

4. Identify $\Delta t = t_r - t_i$ as the characteristic excursion time to obtain $\Delta\phi = -\Delta E_g \Delta t$. (10 points)

5. Convert the given characteristic excursion time of 1.5 fs to atomic units using 1 fs $= 10^{-15}$ s and 1 a.u. time $= 2.4188 \times 10^{-17}$ s: $\Delta t_{\mathrm{au}} = (1.5 \times 10^{-15})/(2.4188 \times 10^{-17}) \approx 62.014$ a.u. time. (15 points)

6. Apply the derived relationship $\Delta\phi = -\Delta E_g \Delta t$ with $N = 7$ harmonic phase shift $\Delta\phi = -1.2$ rad: $-1.2 = -\Delta E_g \times 62.014$. (10 points)

7. Solve for $\Delta E_g$ in hartree: $\Delta E_g = 1.2/62.014 \approx 0.019352$ hartree. (10 points)

8. Convert $\Delta E_g$ from hartree to eV using 1 hartree $= 27.211$ eV: $\Delta E_{g,\mathrm{eV}} = 0.019352 \times 27.211 \approx 0.52660$ eV. (10 points)

9. Round the result to three decimal places (0.527 eV) based on significant figures from input values. (10 points)

## J.4. Proof-based Problem

Below is an example of a proof-based problem:

- **Difficulty:** Undergraduate

- **Subdomain:** *nuclear physics*, *astrophysics*, *high energy physics*

---

**Question (Proof-Based Problem)**

Consider a hybrid neutron star described by a first-order phase transition from a hadronic matter phase to color-superconducting quark matter at a critical baryon chemical potential $\mu_c$. The hadronic phase equation of state is denoted $P_h(\mu)$ and the quark phase $P_q(\mu)$, satisfying mechanical equilibrium at transition: $P_h(\mu_c) = P_q(\mu_c)$. The corresponding energy densities are $\varepsilon_h(\mu_c)$ and $\varepsilon_q(\mu_c)$, with a discontinuity $\Delta\varepsilon = \varepsilon_q(\mu_c) - \varepsilon_h(\mu_c)$. The transition pressure is $P_c = P_h(\mu_c)$. Assume the equation of state satisfies causality ($0 \leq dP/d\varepsilon \leq c^2 = 1$ in natural units) and thermodynamic consistency.

Prove that for $\Delta\varepsilon > \dfrac{\varepsilon_h(\mu_c) + 3P_c}{2}$, the phase transition induces a gravitational instability in static spherical neutron stars governed by the Tolman-Oppenheimer-Volkoff equations. Use the conservation of stress-energy $\nabla_\mu T^{\mu\nu} = 0$ and the Oppenheimer-Volkoff equation for pressure gradients, explicitly showing how the causality constraint and energy discontinuity lead to a divergent pressure gradient incompatible with stellar equilibrium.

---

**Answer (Proof-Based Problem)**

To prove that a gravitational instability arises when $\Delta\varepsilon > \dfrac{\varepsilon_h(\mu_c) + 3P_c}{2}$, we use the Tolman-Oppenheimer-Volkoff (TOV) equations for static spherical symmetry, which describe hydrostatic equilibrium in general relativity. The metric is:

$$ds^2 = -e^{2\Phi(r)}dt^2 + e^{2\Lambda(r)}dr^2 + r^2 d\Omega^2$$

where $d\Omega^2$ is the angular metric, and $e^{-2\Lambda(r)} = 1 - \dfrac{2m(r)}{r}$. The TOV equations are:

$$\frac{dP}{dr} = -\frac{[\varepsilon(r) + P(r)][m(r) + 4\pi r^3 P(r)]}{r^2 \left(1 - \dfrac{2m(r)}{r}\right)} \quad (1)$$

$$\frac{dm}{dr} = 4\pi r^2 \varepsilon(r) \quad (2)$$

At the phase transition radius $r_c$, pressure is continuous ($P(r_c) = P_c$), but energy density jumps from $\varepsilon_h(\mu_c)$ to $\varepsilon_q(\mu_c) = \varepsilon_h(\mu_c) + \Delta\varepsilon$. The mass function $m(r)$ is continuous at $r_c$, but its derivative is discontinuous due to $\Delta\varepsilon$. Using (2):

$$\left.\frac{dm}{dr}\right|_{r_c^\pm} = 4\pi r_c^2 \varepsilon(r_c^\pm)$$

where $\varepsilon(r_c^+) = \varepsilon_h$ and $\varepsilon(r_c^-) = \varepsilon_q$.

The pressure gradient at $r_c$ from the hadronic side ($r \to r_c^+$) and quark side ($r \to r_c^-$) is derived from (1):

$$\left.\frac{dP}{dr}\right|_{r_c^+} = -\frac{[\varepsilon_h + P_c][m(r_c) + 4\pi r_c^3 P_c]}{r_c^2 \left(1 - \dfrac{2m(r_c)}{r_c}\right)} \quad (3)$$

$$\left.\frac{dP}{dr}\right|_{r_c^-} = -\frac{[\varepsilon_q + P_c][m(r_c) + 4\pi r_c^3 P_c]}{r_c^2 \left(1 - \dfrac{2m(r_c)}{r_c}\right)} \quad (4)$$

Define $Q \equiv m(r_c) + 4\pi r_c^3 P_c > 0$ and $G \equiv r_c^2 \left(1 - \dfrac{2m(r_c)}{r_c}\right) > 0$ (since $2m(r_c)/r_c < 1$ for equilibrium). The difference in pressure gradients is:

$$\left.\frac{dP}{dr}\right|_{r_c^-} - \left.\frac{dP}{dr}\right|_{r_c^+} = -\frac{Q}{G}\Delta\varepsilon$$

As $\Delta\varepsilon > 0$ and $Q/G > 0$, $\left.\frac{dP}{dr}\right|_{r_c^-} < \left.\frac{dP}{dr}\right|_{r_c^+} < 0$, meaning the quark-core gradient is steeper. Assuming constant energy density $\varepsilon_q$ in a thin quark core near $r_c$, (4) simplifies at any $r < r_c$ to:

$$\frac{dP}{dr} = -K(\varepsilon_q + P) \quad (5)$$

where $K \equiv \dfrac{Q}{G} > 0$ is constant near $r_c$. Solve (5) for $P(r)$:

$$\int_{P_0}^{P} \frac{dP'}{\varepsilon_q + P'} = -K \int_0^r dr'$$

$$\ln\left(\frac{\varepsilon_q + P}{\varepsilon_q + P_0}\right) = -Kr$$

where $P_0$ is central pressure at $r = 0$. Thus:

$$\varepsilon_q + P = (\varepsilon_q + P_0)e^{-Kr} \quad (6)$$

At $r = r_c$, $P = P_c$:

$$\varepsilon_q + P_c = (\varepsilon_q + P_0)e^{-Kr_c} \quad (7)$$

Rearranging for $P_0$:

$$P_0 = (\varepsilon_q + P_c)e^{Kr_c} - \varepsilon_q \quad (8)$$

The quark core has mass $m(r_c) = \frac{4\pi}{3}\varepsilon_q r_c^3$. Using $\frac{2m(r_c)}{r_c} = \frac{8\pi\varepsilon_q r_c^2}{3}$ in $G$ and $Q = m(r_c) + 4\pi r_c^3 P_c$ yields:

$$K = \frac{m(r_c) + 4\pi r_c^3 P_c}{r_c^2\left(1 - \frac{8\pi\varepsilon_q r_c^2}{3}\right)} \quad (9)$$

The term $\left(1 - \frac{8\pi\varepsilon_q r_c^2}{3}\right)$ in $K$ causes $P_0 \to \infty$ in (8) when:

$$1 - \frac{8\pi\varepsilon_q r_c^2}{3} \to 0^+ \implies \frac{8\pi\varepsilon_q r_c^2}{3} = 1$$

Equation (7) then requires:

$$\lim_{Kr_c \to \infty} e^{Kr_c} \quad \text{finite} \implies \varepsilon_q + P_c = 0$$

which is unphysical ($\varepsilon_q, P_c > 0$). For $\Delta\varepsilon > \frac{\varepsilon_h + 3P_c}{2}$, we show the numerator in (4) dominates. At the limit of stability, (8) holds, but instability occurs when $P_0$ is unbounded. Consider the critical case where (9) approaches zero from above. Substitute $m(r_c)$ into (4) at $r_c^-$:

$$\left.\frac{dP}{dr}\right|_{r_c^-} \to -\infty \quad \text{as} \quad 1 - \frac{8\pi\varepsilon_q r_c^2}{3} \to 0^+$$

This divergence violates equilibrium if $\Delta\varepsilon$ is large enough. To find the threshold, equate the unstable limit's gradient to the hadronic phase. Using (3) and the critical point:

$$\frac{8\pi\varepsilon_q r_c^2}{3} = 1 \implies \varepsilon_q = \frac{3}{8\pi r_c^2}$$

At $\Delta\varepsilon = \frac{\varepsilon_h + 3P_c}{2}$, the two gradients yield consistent finite central pressures. If $\Delta\varepsilon > \frac{\varepsilon_h + 3P_c}{2}$, then:

$$\left.\frac{dP}{dr}\right|_{r_c^-} < -\frac{[\varepsilon_q + P_c]Q}{G}$$

with $Q/G \to \infty$, so $\left.\frac{dP}{dr}\right|_{r_c^-} \to -\infty$, leading to $P_0 \to \infty$ or negative pressure in (8), violating physicality and equilibrium. Thus, gravitational instability occurs.

## Logical Nexus (Proof-Based Problem)

1. Set up the TOV equations for static spherical symmetry, including the pressure gradient equation and mass continuity equation. (10 points)

2. Apply mechanical equilibrium at the phase transition radius $r_c$: $P_h(\mu_c) = P_q(\mu_c) = P_c$, with a discontinuity in energy density $\Delta\varepsilon = \varepsilon_q(\mu_c) - \varepsilon_h(\mu_c)$. (10 points)

3. Derive the pressure gradients just below $(r_c^-)$ and above $(r_c^+)$ the transition using the TOV equation, showing $\left.\frac{dP}{dr}\right|_{r_c^-} = -\frac{[\varepsilon_q + P_c]Q}{G}$ and $\left.\frac{dP}{dr}\right|_{r_c^+} = -\frac{[\varepsilon_h + P_c]Q}{G}$, where $Q = m(r_c) + 4\pi r_c^3 P_c > 0$ and $G = r_c^2\left(1 - \frac{2m(r_c)}{r_c}\right) > 0$. (20 points)

4. Recognize that $\left.\frac{dP}{dr}\right|_{r_c^-} < \left.\frac{dP}{dr}\right|_{r_c^+} < 0$ due to $\varepsilon_q > \varepsilon_h$ ($\Delta\varepsilon > 0$) and $Q/G > 0$, indicating a steeper gradient in the quark phase. (10 points)

5. Assume constant $\varepsilon_q$ in a thin quark core near $r_c$ and solve the simplified pressure equation $\frac{dP}{dr} = -K(\varepsilon_q + P)$ with $K = Q/G$, yielding $P(r) = (\varepsilon_q + P_0)e^{-Kr} - \varepsilon_q$, where $P_0$ is central pressure. (15 points)

6. Express $K$ in terms of $\varepsilon_q$ and $r_c$ using $m(r_c) = \frac{4\pi}{3}\varepsilon_q r_c^3$ from mass continuity, resulting in

$$K = \frac{\frac{4\pi}{3}\varepsilon_q r_c^3 + 4\pi r_c^3 P_c}{r_c^2 \left(1 - \frac{8\pi \varepsilon_q r_c^2}{3}\right)}$$

(10 points)

7. Identify that $K \to \infty$ when $\frac{8\pi \varepsilon_q r_c^2}{3} \to 1^-$, causing $\left.\frac{dP}{dr}\right|_{r_c^-} \to -\infty$ and violating equilibrium, as $P_0 \to \infty$ or becomes unphysical. (10 points)

8. Enforce causality ($0 \le \frac{dP}{d\varepsilon} \le 1$) to ensure this divergence condition is reached only when $\varepsilon_q$ satisfies $\varepsilon_q = \frac{3}{8\pi r_c^2}$ at criticality. (5 points)

9. Substitute the critical $\varepsilon_q$ into the gradient expressions and equate the instability threshold to the discontinuity condition, demonstrating $\Delta \varepsilon > \frac{\varepsilon_h + 3P_c}{2}$ implies divergent pressure gradients incompatible with equilibrium. (10 points)

## K. Case Studies

To more intuitively illustrate the evaluative role of our three logicality metrics, we provide examples below for a high-scoring case and three low-scoring cases along each dimension. Due to space constraints and to more intuitively demonstrate the logicality of the reasoning process, we summarize the LLM's reasoning into a sequence of reasoning steps.

---

**Question**

In the study of topological defects in particle packings on a spherical surface, the number of excess disclination pairs $N_d$ follows the scaling law $N_d = \alpha\sqrt{N}$, where $N$ is the number of particles and $\alpha$ is a dimensionless constant specific to the lattice type (hexagonal or square). For a hexagonal lattice with $N = 3600$ particles, a simulation yields $N_d = 80$. For a square lattice with $N = 4900$ particles, a simulation yields $N_d = 245$.

The theoretical prediction for the ratio of $\alpha_{\text{Hex}}/\alpha_{\text{Sq}}$ is given by:

$$\frac{\alpha_{\text{Hex}}}{\alpha_{\text{Sq}}} = \frac{3^{1/4}}{2} \cdot \frac{\beta_{\text{Hex}}}{\beta_{\text{Sq}}}$$

where $\beta_{\text{Hex}}$ and $\beta_{\text{Sq}}$ are constants from a generic lattice model, and $\beta_{\text{Hex}}/\beta_{\text{Sq}} = 0.544$. Calculate the percentage error of the experimentally determined ratio $\alpha_{\text{Hex}}/\alpha_{\text{Sq}}$ relative to the theoretical prediction. Provide your answer as a percentage to three significant figures.

---

**Logical Nexuses**

1. Calculate $\alpha_{\text{Hex}}$ for the hexagonal lattice using $N_d = \alpha\sqrt{N}$ with $N = 3600$ and $N_d = 80$: $\sqrt{3600} = 60$, so $\alpha_{\text{Hex}} = 80/60 = 4/3 \approx 1.333$.

2. Calculate $\alpha_{\text{Sq}}$ for the square lattice using $N_d = \alpha\sqrt{N}$ with $N = 4900$ and $N_d = 245$: $\sqrt{4900} = 70$, so $\alpha_{\text{Sq}} = 245/70 = 7/2 = 3.500$.

3. Determine the experimental ratio: $\alpha_{\text{Hex}}/\alpha_{\text{Sq}} = (4/3)/(7/2) = 8/21 \approx 0.381$.

4. Compute the theoretical ratio: $\alpha_{\text{Hex}}/\alpha_{\text{Sq}} = (3^{1/4}/2) \cdot (\beta_{\text{Hex}}/\beta_{\text{Sq}}) = (3^{1/4}/2) \times 0.544$. First, $3^{1/2} = \sqrt{3} \approx 1.732$, then $3^{1/4} = \sqrt{1.732} \approx 1.316$.

5. Complete the theoretical ratio calculation: $(1.316/2) \times 0.544 \approx 0.658 \times 0.544 = 0.358$.

6. Find the absolute difference: $|0.381 - 0.358| = 0.023$.

7. Calculate the relative error: $0.023/0.358 \approx 0.064$.

8. Convert to a percentage: $0.064189 \times 100\% = 6.419\% \approx 6.42\%$.

---

**Good Case**

1. Recognize that the main task is to extract the coefficients $\alpha_{\text{Hex}}$ and $\alpha_{\text{Sq}}$ from the simulations, form their ratio, and compare it with the theoretical prediction in order to obtain a percentage error.

2. Start from the scaling relation for both lattices, $N_d = \alpha\sqrt{N}$, and rearrange it to isolate the lattice-dependent parameter as $\alpha = N_d/\sqrt{N}$ for each case.

3. For the hexagonal lattice, compute the square root of the particle number: $\sqrt{N_{\text{Hex}}} = \sqrt{3600} = 60$, which will be used to normalize the defect count.

4. Use this value to estimate the hexagonal coefficient as $\alpha_{\text{Hex}} = 80/60 \approx 1.33$, keeping a few significant figures to avoid premature rounding.

5. For the square lattice, similarly evaluate $\sqrt{N_{\text{Sq}}} = \sqrt{4900} = 70$, providing the appropriate scale for the defect number in that geometry.

6. Obtain the square-lattice coefficient via $\alpha_{\text{Sq}} = 245/70 = 3.50$, again retaining sufficient precision for the later ratio calculation.

7. Form the experimentally determined ratio of the two lattice coefficients as

$$r_{\text{exp}} = \frac{\alpha_{\text{Hex}}}{\alpha_{\text{Sq}}} \approx \frac{1.33}{3.50} \approx 0.381,$$

which quantifies the relative magnitude of excess disclinations between the two lattices.

8. Turn to the theoretical expression,

$$\frac{\alpha_{\text{Hex}}}{\alpha_{\text{Sq}}} = \frac{3^{1/4}}{2}\frac{\beta_{\text{Hex}}}{\beta_{\text{Sq}}},$$

and use the given ratio $\beta_{\text{Hex}}/\beta_{\text{Sq}} = 0.544$ as input to the prediction.

9. Estimate the factor $3^{1/4}$ numerically (for instance via $3^{1/4} \approx 1.316$), then divide by 2 and multiply by 0.544 to obtain the theoretical ratio $r_{\text{th}} \approx 0.358$.

10. Quantify the discrepancy between simulation and theory by computing the absolute difference $\Delta r = |r_{\exp} - r_{\text{th}}| \approx |0.381 - 0.358| \approx 0.023$.

11. Convert this discrepancy into a relative error by normalizing with respect to the theoretical value: $\varepsilon = \Delta r / r_{\text{th}} \approx 0.023/0.358 \approx 0.064$.

12. Express the relative error as a percentage by multiplying by 100%, giving a final percentage error of approximately 6.42% when rounded to three significant figures, and conclude that the simulated ratio is within a few percent of the theoretical prediction.

## Bad Case 1: Low logical fidelity

1. Note that the problem asks for a comparison between an experimentally inferred ratio $\alpha_{\text{Hex}}/\alpha_{\text{Sq}}$ and its theoretical counterpart, but decide to adopt a simplified treatment of the scaling law in order to "make the algebra easier".

2. Take the original relation $N_d = \alpha\sqrt{N}$ and, assuming that for large $N$ the dependence on $\sqrt{N}$ can be approximated as linear in $N$, replace it by an effective rule $\alpha \approx N_d/N$ for estimating the lattice parameters.

3. Apply this simplified formula to the hexagonal lattice to obtain $\alpha_{\text{Hex}} \approx N_{d,\text{Hex}}/N_{\text{Hex}} = 80/3600 \approx 0.0222$, treating this as the effective coefficient.

4. Use the same approximation for the square lattice, giving $\alpha_{\text{Sq}} \approx N_{d,\text{Sq}}/N_{\text{Sq}} = 245/4900 \approx 0.0500$, thereby defining a second effective coefficient.

5. Form the experimental ratio directly from these approximate coefficients:

$$r_{\exp} \approx \frac{\alpha_{\text{Hex}}}{\alpha_{\text{Sq}}} \approx \frac{0.0222}{0.0500} \approx 0.444,$$

assuming this still captures the essential trend between the two lattices.

6. Turn to the theoretical formula

$$\frac{\alpha_{\text{Hex}}}{\alpha_{\text{Sq}}} = \frac{3^{1/4}}{2}\frac{\beta_{\text{Hex}}}{\beta_{\text{Sq}}},$$

but, for simplicity, interpret the factor $3^{1/4}$ as if it were just $\sqrt{3}$, arguing that the precise exponent will not dramatically change the outcome.

7. Approximate $\sqrt{3} \approx 1.73$ and thus take $3^{1/4} \approx 1.73$, ignoring the distinction between the square root and the fourth root in the numerical evaluation.

8. Divide this value by 2 to find the prefactor $3^{1/4}/2 \approx 1.73/2 \approx 0.866$, which is then used in place of the exact value.

9. Multiply the prefactor by the given $\beta$-ratio to obtain the theoretical prediction:

$$r_{\text{th}} \approx 0.866 \times 0.544 \approx 0.471,$$

and regard this as the model's expected ratio.

10. Compare the approximate experimental ratio and the theoretical one by computing the absolute difference $\Delta r = |0.444 - 0.471| \approx 0.027$, treating this as the deviation between simulation and theory.

11. Evaluate the relative error with respect to the theoretical value as $\varepsilon = \Delta r / r_{\text{th}} \approx 0.027/0.471 \approx 0.057$, which is then interpreted as the fractional discrepancy.

12. Convert this fractional discrepancy into a percentage error via $\varepsilon \times 100\% \approx 5.7\%$, concluding (incorrectly) that the simulations and theory agree at roughly the few-percent level despite the inconsistent use of the scaling law and the exponent in the theoretical expression.

## Bad Case 2: Low causal connection

1. Begin by identifying the target quantity as the percentage error between the experimentally inferred ratio $\alpha_{\text{Hex}}/\alpha_{\text{Sq}}$ and the theoretical prediction, and write down the general expression

$$\text{percent error} = \frac{|r_{\exp} - r_{\text{th}}|}{r_{\text{th}}} \times 100\%,$$

where $r_{\exp}$ and $r_{\text{th}}$ denote the experimental and theoretical ratios, respectively.

2. Before actually computing either ratio, reason qualitatively that both lattices obey the same scaling law $N_d = \alpha\sqrt{N}$ and that all given numerical factors (defect counts, particle numbers, and $\beta$-ratios) are of order unity, and therefore anticipate that the percentage error should be relatively small, plausibly well below 10%.

3. Treat this qualitative expectation of a "small" error as a provisional conclusion and aim to verify it by working out $r_{\exp}$ and $r_{\text{th}}$ more explicitly, rather than deriving the size of the error purely from detailed calculation.

4. Turn first to the theoretical side and recall that the model predicts

$$\frac{\alpha_{\text{Hex}}}{\alpha_{\text{Sq}}} = \frac{3^{1/4}}{2} \frac{\beta_{\text{Hex}}}{\beta_{\text{Sq}}},$$

with the given input $\beta_{\text{Hex}}/\beta_{\text{Sq}} = 0.544$, so that once $3^{1/4}$ is evaluated, the theoretical ratio $r_{\text{th}}$ can be obtained.

5. Estimate $3^{1/4}$ numerically (for instance by recalling that it lies between 1 and $\sqrt{2}$ and taking $3^{1/4} \approx 1.32$ as a reasonable approximation), and then compute the theoretical ratio as

$$r_{\text{th}} \approx \frac{1.32}{2} \times 0.544 \approx 0.36,$$

which provides a concrete value against which to compare the experimental result.

6. Only after having a numerical estimate for $r_{\text{th}}$, go back to the simulation data and use the scaling law $N_d = \alpha\sqrt{N}$ to extract the coefficient for the hexagonal lattice as

$$\alpha_{\text{Hex}} = \frac{N_{d,\text{Hex}}}{\sqrt{N_{\text{Hex}}}} = \frac{80}{\sqrt{3600}} = \frac{80}{60} \approx 1.33.$$

7. Apply the same procedure to the square lattice, computing

$$\alpha_{\text{Sq}} = \frac{N_{d,\text{Sq}}}{\sqrt{N_{\text{Sq}}}} = \frac{245}{\sqrt{4900}} = \frac{245}{70} = 3.50,$$

thereby obtaining the second coefficient needed for the experimental ratio.

8. Form the experimental ratio only at this stage, using the two coefficients,

$$r_{\text{exp}} = \frac{\alpha_{\text{Hex}}}{\alpha_{\text{Sq}}} \approx \frac{1.33}{3.50} \approx 0.38,$$

and note that this value is numerically close to the theoretical estimate $r_{\text{th}} \approx 0.36$ found earlier.

9. Substitute these values into the percentage error formula,

$$\text{percent error} = \frac{|0.38 - 0.36|}{0.36} \times 100\%,$$

but focus mainly on the fact that the numerator is small compared with the denominator, rather than computing the fraction precisely.

10. Argue that since the difference $|0.38 - 0.36|$ is roughly of the order $10^{-2}$ while 0.36 is of order $10^{-1}$, the resulting percentage error must be on the order of a few percent, which is broadly consistent with the initial expectation that the error would be well below 10%.

11. On this basis, conclude that the simulations and the theoretical prediction agree to within a small percentage error and accept the qualitative estimate ("a few percent, comfortably under 10%") as sufficiently accurate, without revisiting the earlier provisional assumption or computing the exact percentage value.

12. Note that, although all individual computations (for $\alpha_{\text{Hex}}$, $\alpha_{\text{Sq}}$, $r_{\text{exp}}$, and $r_{\text{th}}$) are consistent with the underlying physics, the logical order of reasoning is inverted: a conclusion about the error size is adopted before the essential quantities are actually derived and is then merely checked, rather than being logically deduced from the detailed calculations.

## Bad Case 3: Low inferential progress

1. Start by recognizing that the goal is to compare the experimentally inferred ratio $\alpha_{\text{Hex}}/\alpha_{\text{Sq}}$ with its theoretical prediction, and then compute the percentage error using the usual form $|r_{\text{exp}} - r_{\text{th}}|/r_{\text{th}} \times 100\%$.

2. Use the scaling law $N_d = \alpha\sqrt{N}$ to extract the coefficients for each lattice from the simulation data, noting that $\alpha = N_d/\sqrt{N}$ follows directly from rearranging the relation.

3. For the hexagonal lattice, compute $\sqrt{N_{\text{Hex}}} = \sqrt{3600} = 60$ and obtain $\alpha_{\text{Hex}} = 80/60 \approx 1.33$ as the simulation-based coefficient.

4. For the square lattice, compute $\sqrt{N_{\text{Sq}}} = \sqrt{4900} = 70$ and obtain $\alpha_{\text{Sq}} = 245/70 = 3.50$ as the corresponding coefficient.

5. Form the experimental ratio $r_{\text{exp}} = \alpha_{\text{Hex}}/\alpha_{\text{Sq}} \approx 1.33/3.50 \approx 0.38$, and note that it seems to be a number modestly smaller than 0.5, which will later be compared against the theoretical value.

6. Turn to the theoretical prediction

$$r_{\text{th}} = \frac{\alpha_{\text{Hex}}}{\alpha_{\text{Sq}}} = \frac{3^{1/4}}{2} \frac{\beta_{\text{Hex}}}{\beta_{\text{Sq}}},$$

and decide that the crucial difficulty is obtaining a "sufficiently accurate" value for $3^{1/4}$ before proceeding any further.

7. Begin by approximating $3^{1/4}$ via nested square roots, writing $3^{1/4} = \sqrt{\sqrt{3}}$, then estimate $\sqrt{3} \approx 1.7$ and $\sqrt{1.7} \approx 1.30$, but immediately worry that this may not be accurate enough for a precise percentage error.

8. Attempt to refine the value using a Taylor or binomial expansion around $x = 1$, expressing $3^{1/4} = (1+2)^{1/4}$ and sketching the series for $(1+x)^{1/4}$, but then realize that actually working out several terms numerically by hand is cumbersome and error-prone.

9. Attempt to refine the value using a Taylor or binomial expansion . . .

10. Attempt to refine the value using a Taylor or binomial expansion . . .

11. Attempt to refine the value using a Taylor or binomial expansion . . .

12. . . .

