# OpenReview forum: "Scientific logicality enriched methodology for LLM reasoning: A practice in physics"
_ICML.cc/2026/Conference — ICML 2026 regular_

### Official Review · Reviewer_qDxA · 2026-03-10

**Soundness:** 3
**Presentation:** 3
**Significance:** 2
**Originality:** 2
**Overall Recommendation:** 3
**Confidence:** 3

**Summary:**

This paper introduce a new dataset and a benchmark in theory-centric Physical question answering. Logical reasoning steps are explicitly emphasized in the dataset and benchmark. A series of metrics are proposed to evaluate the reasoning steps. Finetuning model with specific size (~7B) on the proposed dataset sees the improvement on the proposed benchmark and some other benchmarks.

**Compliance With Llm Reviewing Policy:**

Affirmed.

**Final Justification:**

After the rebuttal, most of my concerns on experiments are addressed. I sill have concerns on the motivation and soundness of the work. As a result, I will raise my score to 3. Unfortunately, I cannot champion for the acceptance of this paper.

**Key Questions For Authors:**

See above.

**Limitations:**

Yes

**Strengths And Weaknesses:**

Strengths:
1. The metrics can be useful for the evaluation of LLM reasoning.
2. The experiments are extensive.

Weaknesses:
1. The motivation of proposing such a dataset as well as the benchmark is unclear. For the benchmark, why it is necessary to introduce logical reasoning steps? In the end, the reasoning processes of LLM and human are not necessary to be identical. For the dataset, I think most of the recent advanced LLMs have included the arXiv papers used to construct the dataset in their pre-taining/post-training.
2. As a benchmark/dataset paper, scalability is very important. The benchmark is limited in size. And the experiments only involve LLMs with small parameter size.
3. The relation and difference between Logic-Distill and RST is unclear. Why do we need to construct two datasets. And when we want to fine-tune an LLM, which dataset should we choose?
4. The empirical effectiveness is questionable. In-domain performance is good, but Table 6 shows that both Logic-Distill and RST cannot stably improve the model performance. On Math domain datasets AIME2025 and AMC, which are much more different from the training set than those benchmarks in Table 6, the RST performance is worse. In the construction of dataset, the author already focused on theory-centric Physics papers, and the appendix show that the constructed questions are very *mathematic*. Even in this case, the generalization to mathematical questions seems to be weak.

---

> ### Author Rebuttal · Authors · 2026-03-28
>
> We sincerely thank the reviewer for the careful reading and constructive comments. Below, we provide detailed responses to the raised weaknesses.
>
> ### **Response to Weakness #1**
>
> Thank you for your comment. Rather than assessing whether LLM reasoning resembles human reasoning, this work focuses on the effectiveness of reasoning steps and the reliability of conclusions in scientific reasoning, namely, logicality. Although recent reasoning LLMs such as DeepSeek-R1 and OpenAI o1 can solve complex problems with long reasoning chains, their traces often involve lengthy iterations with limited logical coherence. Moreover, Section 4 shows that higher logicality is closely associated with better reasoning performance, highlighting the practical importance of studying logicality in LLM scientific reasoning.
>
> Since academic papers naturally encode rigorous deductive chains, we construct both the benchmark and training dataset from academic papers. To improve benchmark objectivity and fairness, we select papers released after 2025, which helps reduce the risk of test leakage. Furthermore, the proposed data construction methodology and framework are highly reusable. In future work, we will continue incorporating newly published papers and updating the benchmark to support more open and objective evaluation for advanced LLMs.
>
> ### **Response to Weakness #2**
>
> Thank you for raising this question. As described in Section 3, the proposed PHYSLOGIC benchmark contains 864 carefully curated problems, covering 9 physics subfields, 4 difficulty levels, and 4 problem types, with each difficulty–type combination balanced by design. This gives PHYSLOGIC the coverage, balance, and diagnostic value needed for a systematic evaluation of LLM scientific logicality in physics. In terms of scale, PHYSLOGIC is also comparable to existing widely used scientific QA benchmarks, such as GPQA-main (448), SciBench (692), PhysReason (1200), and AMC23 (40).
>
> For the training experiments, we use three medium-scale (~8B) open-source LLMs as backbones, mainly for controllability, reproducibility, and computational budget. We will extend the experiments to larger-scale backbones in future work to further explore the upper bound of LLM capabilities in scientific reasoning with strong logicality.
>
> ### **Response to Weakness #3**
>
> Thank you for raising this question. *Logic-Distill* and *RST* are two complementary supervision methods for scientific logicality. They share the same goal, but differ in supervision granularity and application scenarios. *RST* directly organizes logical nexuses into coherent reasoning processes, providing a more abstract and concise supervision signal. In contrast, *Logic-Distill* uses LLM-generated reasoning and treats logicality as an indirect filtering signal, thereby preserving more explicit intermediate steps and local logical connections as finer-grained supervision.
>
> The results in Table 6 further reflect these differences. *RST* is generally more effective for instruction-tuned LLMs with stronger instruction-following and reasoning-format alignment, but less stable than distillation-based methods with longer reasoning chains and finer-grained supervision for base LLMs. Across tasks, *RST* works better on multiple-choice benchmarks such as GPQA, while distillation-based methods perform better on computational tasks such as SciBench and PhysReason. We will add further discussion of this point in the revised paper.
>
> ### **Response to Weakness #4**
>
> Thank you for your comment. In Table 6, when instruction-tuned models are used as backbones, both *RST* and *Logic-Distill* outperform all baselines overall. On Qwen-7B-Instruct, they surpass the strongest baseline by 5.12% and 8.75%, respectively; on DeepSeek-R1-7B, the gains are 4.92% and 6.08%. For Llama-3.1-8B, due to stronger scaling-law effects on base LLMs, 40k *Logic-Distill* is weaker than 80k *Direct-Distill*. Still, under the same 40k budget, *Logic-Distill* achieves SOTA performance (see Figure 4 (2-a)), as discussed in Section 4.4. Overall, these results provide solid empirical support for the effectiveness of our logicality-enhanced training data on public benchmarks.
>
> Regarding the mathematical benchmarks, although our training data contains substantial mathematical derivations, it is still drawn from theory-centric physics papers and supports physics-specific scientific reasoning rather than competition-style mathematical problem solving. Thus, mathematical intensity does not imply the same task distribution as AIME or AMC. Moreover, as discussed in Response to Weakness #3, *RST* provides a more abstract and concise supervision signal, whereas *Logic-Distill* preserves more complete intermediate computations, making it better suited to mathematical benchmarks. Overall, our work focuses on physics-domain scientific reasoning, where the method has shown strong empirical effectiveness. We will extend this line of work to other disciplines in future work.

---

> > ### Author Rebuttal · Reviewer_qDxA · 2026-04-03
> >
> > 1. I cannot understand the explanation *this work focuses on the effectiveness of reasoning steps and the reliability of conclusions in scientific reasoning, namely, logicality*. I mean the reasoning steps of LLMs can be different from a human being when solving a same problem. If the LLMs can solve the problem, why caring about the reasoning steps? Besides, please show me the evidence that the LLMs' reasoning *traces often involve lengthy iterations with limited logical coherence*.
> > 2. Without actual results, the scalability is still unknown.
> > 4. If the method is not proved to be generalizable, let us look into the framework `Constructing datasets in a domain -> train a model -> evaluate the performance on the exact domain`. How do you know the improvement is not due to the strong correlation between data labeling and evaluation process?

---

> > > ### Author Response · Authors · 2026-04-04
> > >
> > > Thanks for your comment and sorry for the confusion. Below, we provide detailed responses to the follow-up questions.
> > >
> > > ### **Response to follow-up Q.1**
> > >
> > > First, our **understanding of scientific reasoning** is based on the research findings from cognitive and educational psychology (Díaz et al, 2023; Fischer et al, 2014), indicating that the scientific reasoning process in a discipline encompasses a set of interrelated logical/epistemic activities (i.e. logicality) to ensure the validity of reasoning steps and the reliability of conclusions. Thus, logicality reflects the inherent reasoning paradigm for solving problems within a scientific discipline, which is generally held in common by its practitioners presented in previous study (Dowden, 1993).
> > >
> > > Based on these theoretical groundings, logicality can be viewed as the inferential property and reasoning formulation of its own in this discipline (that can be learned or practiced by professional human but *not* of human), to improve the effectiveness of reasoning steps leading to reliable performances for intelligent entities (regardless of a *human* or *LLMs*). In constructing the benchmark and dataset, while it is true that we utilize scientific papers written by human experts, our focus is essentially on identifying the **internal logicality commonly shared** within the scientific community, rather than particularly aligning to human reasoning process (the illustration in *Figure 1* aims similarly). Empirical results in Section 4.2 further support that higher logicality is closely associated with better reasoning performance, highlighting the **advantage of our logicality-centric methodology in LLM physics reasoning**.
> > >
> > > For evidence of LLMs reasoning traces, *DeltaBench* study across multiple reasoning LLMs by He et al (ACL 2025) reports averagely 67.8% reflections are ineffective and 27% reasoning sections are redundant. *LCoT2Tree* (Jiang et al, EMNLP25) and *ConCISE* (Qiao et al, EMNLP25) reveal similar problems of lengthy iterations as illustrated in *Figure 1* with DeepSeek R1 in our paper.
> > >
> > > ### **Response to follow-up Q.2**
> > >
> > > We conducted experiments utilizing the larger-scale Qwen-2.5-14B-Instruct as backbone, with MegaScience and Direct-Distill (DD) serving as comparative baselines, and evaluated the results on PhysLogic, GPQA, and SciBench.
> > >
> > > | Datasets    | Avg. Logicality | GPQA      | SciBench  | PhysLogic | Avg. Accuracy |
> > > | ----------- | --------------- | --------- | --------- | --------- | ------------- |
> > > | MegaScience | 43.34           | 33.72     | 41.45     | 40.05     | 38.41         |
> > > | DD          | 44.60           | 29.07     | 47.15     | 43.06     | 39.76         |
> > > | **LD**      | 47.30           | 38.37     | **50.26** | **47.22** | **45.28**     |
> > > | **RST**     | **49.85**       | **43.02** | 47.15     | 45.60     | 45.26         |
> > >
> > > In the table above, based on a larger LLM, both of our proposed methods, LD and RST, outperform the baseline methods in terms of both scientific logicality and final performance. **These actual results provide strong evidence for the scalability of our methods.** We will add these experiments to the revised paper.
> > >
> > > ### **Response to follow-up Q.3**
> > >
> > > In our main experiments, we construct the training dataset in the physics domain and evaluate the model within the same domain. This experimental paradigm is also adopted by most of the existing studies on scientific reasoning (Zheng et al, *PHYSICS* dataset, NIPS25; Yu et al, *MetaMath*, ICLR24). To isolate the effect of scientific logicality from data labeling, we use rejection sampling (see *Dataset Construction* paragraph) to ensure that, for each training question, the final answers under *Direct-Distill (DD)*, *Logic-Distill (LD)*, and *RST* are all correct and identical, differing only in their reasoning processes. Therefore, **under consistent labeling**, the superior performance of *RST* and *LD* over *DD* in Table 5 and Table 6 **can be attributed to improvements in reasoning logicality**.
> > >
> > > Furthermore, regarding your concern that our method may have limited generalization in mathematics, we have conducted more systematic experiments in the math domain, including the addition of a new baseline and two more math benchmarks.
> > >
> > > | Datasets    | math500   | gsm8k     | AIME25   | AMC       | Avg.      |
> > > | ----------- | --------- | --------- | -------- | --------- | --------- |
> > > | MegaScience | 72.40     | 86.73     | 6.79     | 40.31     | 51.56     |
> > > | DD          | 71.20     | 84.38     | 6.25     | 41.19     | 50.76     |
> > > | **LD**      | **76.00** | **90.45** | **8.54** | **43.15** | **54.54** |
> > > | **RST**     | 74.60     | 88.70     | 8.13     | 40.81     | 53.06     |
> > >
> > > As shown above, *LD* achieves the best performance on all four benchmarks, while *RST* consistently outperforms the baselines on average. These results provide further evidence that **our methods generalize well under a cross-domain setting**. We will include these in the revised paper.

---

### Official Review · Reviewer_Zrfx · 2026-03-12

**Soundness:** 3
**Presentation:** 3
**Significance:** 3
**Originality:** 3
**Overall Recommendation:** 4
**Confidence:** 3

**Summary:**

This paper studies how to improve scientific reasoning in Large Language Models (LLMs) by focusing not just on task performance but on the logicality of the reasoning process. The authors argue that current approaches mostly train LLMs to answer scientific questions by providing more data or longer reasoning chains, but they often overlook whether the reasoning steps themselves are logically sound and coherent.

To address this, the paper makes three main contributions:

1. Scientific Logicality Metrics: The authors define three metrics—F (Logical Fidelity), O (Causal Connection), and P (Inferential Progress)—to quantitatively measure the logical quality of an LLM’s reasoning steps. F evaluates content alignment with core logical steps, O evaluates whether the reasoning maintains correct causal order, and P evaluates whether the reasoning progresses forward without redundant loops.

2. Logic-Enriched Data Sampling Methods for SFT: They propose two methods to generate high-quality fine-tuning data:

Logic-Distill: filters LLM-generated reasoning using F/O/P scores to retain only high-logicality examples.

Reasoning Style Transfer (RST): converts core logical steps into natural, human-readable reasoning paths while preserving logical structure.
These methods aim to train models not just to give correct answers but to reason logically.

3. Physics Benchmark and Dataset Construction: Using physics as an exemplar domain, they construct an 80k-instance training dataset and the PHYSLOGIC benchmark (864 questions) derived from research papers, ensuring a wide coverage of problem types, difficulty levels, and logical reasoning steps.

Key Findings:

1. Models trained with Logic-Distill and RST data achieve higher logicality (F/O/P scores) and better final-answer accuracy than baseline methods.

2. Filtering or transforming data for logicality is more effective than simply increasing dataset size.

3. Ablation studies confirm that each metric (F, O, P) contributes to improved reasoning performance.

Overall, the paper demonstrates that enhancing logical fidelity in LLM training data improves both reasoning quality and task performance, and provides a systematic methodology and benchmark for evaluating scientific logicality in physics reasoning.

**Compliance With Llm Reviewing Policy:**

Affirmed.

**Final Justification:**

The paper has no major issues overall, and I therefore support a Weak Accept.

**Key Questions For Authors:**

1. Difficulty levels in the physics dataset: Physics papers often focus on research-level questions, yet the dataset includes HS, UG, and Grad-level questions. Could the authors clarify why lower-difficulty questions are included, and whether there exist any research-level questions in the dataset?

2. Out-of-domain performance of RST: RST shows limited and inconsistent improvements in out-of-domain experiments. Could the authors provide insights into the cause and potential strategies to improve generalization?

3. Rigorous definition of F, O, P metrics: Are there more principled or reproducible ways to define and compute F, O, and P metrics beyond the current heuristic scoring?

**Limitations:**

yes

**Strengths And Weaknesses:**

Strengths

1. Clear motivation and technical soundness: The paper tackles an important problem in LLM reasoning, is technically solid, and conducts thorough experiments.

2. Well-structured presentation: The narrative is clear, with intuitive and visually understandable results.

3. Significant and valuable research problem: The work addresses an interesting question and provides a useful dataset for the community.

4. Novelty in logicality-aware metrics and data construction: The introduction of F, O, P metrics and the logic-guided dataset construction show creative thinking and contribute new tools for reasoning evaluation.

Weaknesses

1. Heuristic nature of metrics: The definitions of F, O, and P rely on heuristic scoring and lack a more rigorous and reliable computational method.

2. Limited out-of-domain generalization: RST improvements in out-of-domain experiments are small and inconsistent, raising concerns about generalizability.

---

> ### Author Rebuttal · Authors · 2026-03-28
>
> We sincerely thank the reviewer for the careful reading and constructive comments. Below, we provide detailed responses to the raised weaknesses and questions.
>
> ### **Response to Weakness #1**
>
> Thank you for your comment. We agree that, for open-ended scientific reasoning, existing research still lacks a widely accepted and directly computable formal definition of “logicality”. Therefore, in this work, we operationalize scientific logicality into three complementary metrics, namely $\mathcal{F}$, $\mathcal{O}$, and $\mathcal{P}$, which capture the logicality of content coverage, logical ordering, and inferential progress, respectively. More importantly, in Section 4.2, we conduct substantial empirical studies to examine whether our proposed metrics can genuinely capture the logicality of reasoning. The results show that our automatic logicality metrics align well with both human and LLM judgments, and that higher logicality scores are closely associated with better reasoning performance. These findings suggest that the three proposed metrics can serve as a reasonable, stable, and effective evaluation signal for the logicality of LLM reasoning. In future work, we will further explore more robust and interpretable evaluation criteria for the logicality of scientific reasoning.
>
> ### **Response to Weakness #2**
>
> Thank you for pointing this out. In the in-domain experiments, to avoid methodological circularity, we only evaluated *RST*, which shows clear advantages over the baselines. In the out-of-domain experiments, we evaluated both *RST* and *Logic-Distill*. As shown in Table 6, with the two instruction-tuned backbones, Qwen2.5-7B-Instruct and DeepSeek-R1-Distill-Qwen-7B, *RST* outperforms all baselines and is overall second only to *Logic-Distill*; on GPQA, it even performs better. By contrast, *RST* performs less favorably with the base model Llama-3.1-8B.
>
> We attribute this to the nature of *RST*, which mainly transfers a high-logicality reasoning style and thus tends to produce shorter, more concise reasoning. This type of supervision is more effective for instruction-tuned models with stronger instruction-following ability and reasoning-format alignment, but less stable for base models than distillation methods with finer-grained supervision and longer reasoning chains. A similar pattern appears across task types: *RST* works better on multiple-choice tasks such as GPQA, while distillation-based methods are stronger on computational tasks such as SciBench and PhysReason.
>
> Therefore, these results do not indicate a failure of *RST* in generalization; rather, they suggest that different logicality-enhanced methods have different strengths across backbones and task types. In future work, we will explore hybrid approaches that combine reasoning-style transfer with distillation to improve generalization across settings.
>
> ### **Response to Question #1**
>
> Thank you for your question. The PHYSLOGIC benchmark proposed in this paper is constructed from physics papers and covers four difficulty levels, ranging from high school to PhD. However, even the lower-difficulty questions are not simple factual judgments; they still require multi-step logical reasoning and calculation. The difference in difficulty mainly reflects differences in prerequisite knowledge, rather than a lack of logical depth in the reasoning process itself. As described in Section 3, the Master’s- and PhD-level questions are closer to academic research settings, and often require systematic professional training as well as frontier knowledge in specific research directions to be answered. In the revised paper, we will further clarify this point for better presentation.
>
> ### **Response to Question #2**
>
> Thank you for your question. As noted in our response to Weakness #2 above, the limited and unstable out-of-domain gains of *RST* are mainly due to its emphasis on transferring a high-logicality reasoning style, which usually leads to shorter and more concise responses. This type of supervision is more effective for instruction-tuned models with stronger instruction-following ability and better alignment with reasoning formats, and it also works better on multiple-choice tasks such as GPQA, where intuitive logical deduction plays a larger role. We will clarify this more explicitly in the revised paper. In future work, we will explore hybrid approaches that combine reasoning-style transfer with fine-grained distillation to improve generalization across settings.
>
> ### **Response to Question #3**
>
> Thanks. We agree that exploring more principled and reproducible definitions of evaluation metrics will be an important part of our future work, for example, through symbolic parsing, equation-level verification, or graph-based reasoning alignment. We will add a discussion of these future directions in the revised paper.

---

> > ### Author Rebuttal · Reviewer_Zrfx · 2026-04-01
> >
> > I have no further questions. I still consider a score of 4 (Weak Accept) appropriate and will keep my rating unchanged.

---

> > > ### Author Response · Authors · 2026-04-01
> > >
> > > Dear Reviewer Zrfx, Thank you very much for your time and positive feedback! We are glad that our rebuttal and clarifications have addressed all your concerns.

---

### Official Review · Reviewer_7nP1 · 2026-03-13

**Soundness:** 3
**Presentation:** 3
**Significance:** 3
**Originality:** 2
**Overall Recommendation:** 4
**Confidence:** 3

**Summary:**

This paper studies the logical structure of scientific reasoning in LLMs. The authors propose a framework named Scientific Logicality Enriched Methodology to evaluate and improve the logical quality of model reasoning. They introduce three metrics that measure logical fidelity, causal connection, and inferential progress. Based on these metrics, the paper designs logic guided sampling strategies to construct SFT training data from physics papers. Experiments on the proposed benchmark and several public physics datasets show that training with logically structured data improves both reasoning quality and final task accuracy.

**Compliance With Llm Reviewing Policy:**

Affirmed.

**Final Justification:**

My concerns are mainly answered. I will maintain my positive score.

**Key Questions For Authors:**

Q1. How sensitive are the proposed logicality metrics to errors in the extracted logical nexuses?

Q2. Have the authors considered using symbolic reasoning tools or equation parsers instead of sentence embeddings for evaluating logical relations?

Q3. How does the method perform on problems that require long mathematical derivations rather than short reasoning chains?

Q4. Is the framework applicable to multimodal scientific reasoning that involves figures, diagrams, or experimental data?

**Limitations:**

First, the proposed evaluation relies on logical nexuses extracted by an LLM, which may introduce noise and bias into both evaluation and training. Second, the current framework evaluates reasoning primarily in text-based QA tasks and does not yet address more complex scientific workflows or multimodal reasoning scenarios.

**Strengths And Weaknesses:**

#### Strengths

- The paper proposes a clear framework. It introduces explicit criteria to evaluate scientific logicality and connects evaluation with data construction and training. The overall pipeline is coherent and technically sound.

- The authors build training data using logic guided sampling strategies. Experiments show that training on such data improves reasoning quality and task accuracy. This provides useful evidence that logically structured supervision can benefit scientific reasoning.

- The ablation experiments indicate that maintaining the correct causal order of reasoning steps plays a critical role. This result highlights the importance of reasoning structure rather than only increasing reasoning length.

#### Weaknesses

- W1. The logical nodes used as ground truth are generated by an LLM from papers. This process may introduce bias or errors. If the extracted structure is incorrect, the evaluation metrics and training signals may also become unreliable.

- W2. The proposed metrics rely on sentence embeddings and cosine similarity. However, scientific reasoning often involves symbolic derivation and mathematical formulas. Pure semantic embeddings may not accurately capture the logical relations in such reasoning processes.

- W3. Although the dataset is constructed from scientific papers, the final tasks are still QA problems. They do not cover more complex scientific workflows such as hypothesis generation, experiment design, or multi-stage reasoning. In addition, reasoning is limited to text and does not consider multimodal scientific information.

---

> ### Author Rebuttal · Authors · 2026-03-28
>
> We sincerely thank the reviewer for the careful reading and constructive comments. Below, we provide detailed responses to the raised weaknesses and questions.
>
> ### **Response to Weakness #1**
>
> Thanks. To reduce the bias or errors potentially introduced by LLM-based extraction of logical nexuses, we applied a three-stage quality control pipeline, including rule-based filtering, LLM-based filtering, and human evaluation, rather than directly using the extracted results (see Appendix E.2). In human evaluation, two experts gave the dataset an average Nexus Quality score above 90 with good agreement (see Table 19). Moreover, the logicality assessment shows good alignment with human/third-party ratings (see Section 4.2), supporting the extracted nexuses as reasonably reliable structured supervision signals for evaluation and data selection.
>
> ### **Response to Weakness #2**
>
> We thank the reviewer for the comment. In our framework, sentence embeddings are used only as a low-level representation tool to map each local reasoning step and logical nexus into a shared vector space, providing semantic relatedness signals rather than directly modeling logical relations. Based on these signals, we construct three higher-level metrics, $\mathcal{F}$, $\mathcal{O}$, and $\mathcal{P}$, to assess logicality from complementary perspectives, including content alignment, step connection/order, and inferential progress. Thus, cosine similarity is not treated as logicality itself, but only as an input to higher-level logicality assessment. The good alignment with human/third-party ratings in Section 4.2 further supports the effectiveness of this design for our task setting.
>
> ### **Response to Weakness #3**
>
> We thank the reviewer for the constructive suggestion. The dataset in this work is formulated in a QA format mainly because such tasks usually have relatively well-defined and objectively verifiable answers, which provide a more unified and controlled evaluation interface across different question types, difficulty levels, and models. At the same time, our benchmark does not only assess the final answer, but further evaluates the logicality of the reasoning process, and therefore is not equivalent to a traditional QA benchmark. We also agree with the reviewer’s suggestion regarding more complex scientific workflows and multimodal scientific information. In future work, we will further extend our setting to more complex tasks such as hypothesis generation and experiment design, and explore the construction of corresponding multimodal training and benchmark datasets.
>
> ### **Response to Question #1**
>
> Thank you for the question. Our metrics are based on weighted aggregation and structural evaluation over multiple logical nexuses, rather than relying on any single nexus, so local extraction errors are more likely to introduce limited noise rather than systematic distortion. In addition, as shown in Section 4.2, the good alignment between our metrics and human/third-party ratings empirically supports their stability under a certain level of nexus noise. We also thank the reviewer for this suggestion, and in future work we will further study the impact of logical nexuses on the overall evaluation metrics and explore ways to improve the robustness of the logicality assessment framework.
>
> ### **Response to Question #2**
>
> Thank you for the suggestion. Symbolic reasoning tools or equation parsers are indeed a valuable direction, especially for formula-intensive reasoning. In this work, we use sentence embeddings as a unified and scalable representation because our benchmark includes both symbolic and natural-language scientific reasoning. We will explore integrating symbolic or formula-aware methods in future work.
>
> ### **Response to Question #3**
>
> Thank you for the question. Our experiments are not limited to short reasoning chains. In the in-domain setting, PHYSLOGIC already includes computation and proof problems derived from core physics derivations, and Table 1 shows that the average reasoning length in several categories is close to 10,000 tokens. In the out-of-domain setting, we further evaluate on PhysReason, a public physics reasoning benchmark with competition-style computational problems and step-level verification. This suggests that our method can also apply to problems requiring relatively long derivations.
>
> ### **Response to Question #4**
>
> Thank you for the question. Our current framework is developed in a text-based setting, but it is in principle extensible to multimodal scientific reasoning. We view this as an important future direction, and will further explore extending the logicality assessment framework to multimodal scientific scenarios.
>
> ### **Response to Limitations**
>
> We thank the reviewer for highlighting these limitations. We have clarified these related issues in the rebuttal above. We will further elaborate on these limitations in the revised paper and investigate them more thoroughly in future work.

---

> > ### Author Rebuttal · Reviewer_7nP1 · 2026-04-02
> >
> > Thank you for the detailed rebuttal. I will maintain my positive score.

---

> > > ### Author Response · Authors · 2026-04-02
> > >
> > > Dear Reviewer 7nP1, Thank you very much for your time and positive feedback! We are glad that our rebuttal and clarifications have addressed all your concerns.

---

### Decision · Program_Chairs · 2026-04-30

**Decision:**

Accept (regular)

**Comment:**

This is an interesting paper that introduces criteria to evaluate scientific logicality and connects it with data construction and model training. The data synthesis pipeline is technically sound relying on logic guided approaches. The paper then shows that training on such data improves scientific reasoning. The reviewers like the technical exposition and usefulness of the approach, although some also questioned how it can work at scale for a broad categories of tasks. Yet, I think this is an interesting contribution.